# Dysplastic lung repair fosters a tuberculosis-promoting microenvironment through maladaptive macrophage polarization

Shivraj M. Yabaji[1], Suruchi Lata[1], Anna E. Tseng[1], Prasanna Babu Araveti[1], Ming Lo[1], Igor Gavrish[1], Aoife K. O'Connell[1], Hans P. Gertje[1], Anna C. Belkina[2,3,4], Colleen E. Thurman[1], Hirofumi Kiyokawa[2,5], Darrell Kotton[2,5], Shumin Tan[6], Janice J. Endsley[7], William R. Bishai[8], Nicholas Crossland[1,3], Lester Kobzik[9], Igor Kramnik[1,2]*

1 The National Emerging Infectious Diseases Laboratories (NEIDL), Boston University, Boston, Massachusetts, United States of America, 2 Pulmonary Center, The Department of Medicine, Boston University Chobanian & Aveedisian School of Medicine, Boston, Massachusetts, United States of America, 3 The Department of Pathology and Laboratory Medicine, Boston University Chobanian & Avedisian School of Medicine, Boston, Massachusetts, United States of America, 4 Flow Cytometry Core Facility, Boston University School of Medicine, Boston, Massachusetts, United States of America, 5 Center for Regenerative Medicine of Boston University and Boston Medical Center, Boston, Massachusetts, United States of America, 6 Department of Molecular Biology and Microbiology, Tufts University School of Medicine, Boston, Massachusetts, United States of America, 7 Departments of Microbiology & Immunology and Pathology, The University of Texas Medical Branch, Galveston, Texas, United States of America, 8 Center for Tuberculosis Research School of Medicine, John Hopkins University, Baltimore, Maryland, United States of America, 9 Cellecta, Inc., Mountain View, California, United States of America

* ikramnik@bu.edu

## Abstract

Pulmonary TB that develops in immunocompetent adult humans is responsible for approximately 85% of the disease burden and is central for Mtb transmission. Most humans contain Mtb infection within primary granulomatous lesions, but in certain immunocompetent humans, containment fails, leading to hematogenous spread and active pulmonary disease with the formation of cavities that enable Mtb transmission via aerosols. To reveal lung-specific microenvironments conducive for Mtb survival and replication despite systemic immunity, we use fluorescence multiplex immuno-histochemistry and spatial transcriptomic analyses of heterogenous TB lesions that uniquely form in the lungs of immunocompetent but TB-susceptible B6.Sst1S mice after hematogenous spread from the primary lesion. Initially, these secondary lung lesions manifested local adoptive immunity featuring tertiary lymphoid follicles similar to resistant B6 mice and contained primarily non-replicating bacilli. Following these early events, however, the B6.Sst1S mice uniquely demonstrate expansion of myeloid cell populations with the appearance of alternatively activated macrophages, dissolution of lymphoid follicles, and the accumulation of de-differentiated lung epithelial cells. These processes led to bronchogenic expansion, broncho-occlusion, and necrosuppurative pneumonia closely resembling advanced pulmonary tuberculosis in humans. To determine whether lung parenchymal cells or lung oxygenation were necessary for

**Data availability statement:** The Spatial transcriptomics data was deposited to GEO dataset GSE292392 (https://www.ncbi.nlm.nih.gov/geo/query/acc.cgi?acc=GSE292392).

**Funding:** This work was supported by National Institutes of Health grant R01HL126066 (IK). The funders had no role in study design, data collection and analysis, decision to publish, or preparation of the manuscript.

**Competing interests:** The authors declare that there are no competing interests.

the pulmonary TB progression, we implanted lung and spleen fragments subcutaneously prior to the infection. The lung implants uniquely displayed the formation of the characteristic organized granulomas with necrosis and Mtb replication that paralleled TB progression in native lungs, demonstrating that the cellular composition of inflamed lung tissue, not oxygenation, is a critical determinant of pulmonary TB progression. Our data demonstrate that deleterious bi-directional interactions of aberrantly activated macrophages with the inflammation-injured lung resident cells determine lung vulnerability to virulent Mtb in immunocompetent hosts. Because these mechanisms enable Mtb transmission among humans via aerosols, they are likely evolutionary conserved and, therefore, represent appealing targets for host-directed TB therapies.

### Author summary

Pulmonary tuberculosis (PTB) represents 85% of the disease burden caused by *Mycobacterium tuberculosis* (Mtb). To study mechanisms that favor Mtb replication in the lungs despite systemic immunity, we developed a mouse model resembling the PTB of humans. We revealed an expansion of dysplastic lung epithelial cells and alternatively activated Arg1+ macrophages (AAM) in PTB lesions and discovered M2 polarization in peripheral blood cells. To explore the roles of parenchymal cells and oxygenation in the AAM recruitment to the lung, we implanted lung and spleen fragments subcutaneously. After Mtb infection, the lung, but not spleen, implants recruited AAMs and displayed characteristic necrotic granulomas and Mtb replication. Necrotic TB lesions also developed in subcutaneous implants of human lung tissue in mice with a human immune system. The results demonstrate that deleterious interactions of aberrantly activated macrophages with the inflammation-injured lung resident cells, are critical determinants of PTB progression in immunocompetent hosts.

### Introduction

Tuberculosis remains a top global health problem. It is estimated that ~25% of all people have been infected with *Mycobacterium tuberculosis* (Mtb) [1]. Most immunocompetent humans, however, restrict primary TB infection and remain non-infectious. However, approximately 5–10% of Mtb-infected people can later develop active TB disease during their lifetime. Pulmonary TB (PTB) is the major clinical form of TB that accounts for approximately 85% of all TB cases representing the major causes of TB morbidity and mortality [1–3]. In endemic areas primary infection with Mtb usually occurs in childhood, while the development of the most severe disease form - pulmonary TB with cavitary lesions - occurs in immunocompetent adults. How the later clinical form develops despite prominent Mtb-specific immune responses, and why PTB is not efficiently prevented by BCG immunization remains insufficiently understood [4–6]. Answering these questions is essential for the development of novel, more efficient preventive and therapeutic interventions.

Mtb is a specialized human pathogen that does not have natural reservoirs, and its co-evolution with natural human hosts is entirely dependent on human-to-human transmission [7]. The unique vulnerability of the human lung to TB is crucial for Mtb transmission via aerosols - the only epidemiologically significant route in human populations. Lung tissue serves as a gateway for Mtb infection, where finely aerosolized Mtb particles reach alveoli to infect alveolar macrophages and induce the formation of primary TB lesions [8]. These lesions are often sufficient to contain the infection and "arrest" the bacterial spread without complete eradication, thus setting a stage for subsequent TB progression, when conditions permit. The formation of cavitary pulmonary TB formation is an end stage of Mtb life cycle within the host that is required to produce infectious aerosols [6]. An improved understanding of the mechanisms behind failed TB containment uniquely in the lungs of immunocompetent individuals would enable interventions to control the PTB progression and transmission among natural human hosts. However, its mechanisms are much less understood and studied as compared to primary TB [4,9].

Current evidence suggests that pulmonary TB in previously infected individuals may develop after respiratory re-infection [10,11] or as shown in classic studies after seeding the lungs following hematogenous spread from primary lesions(reviewed in [3,6,12,13]). In animal models, bacterial dissemination occurs during the initial weeks after the aerosol infection [14–16]. In humans, there is also a stage of detectable Mtb bacteremia [17] that may occur within weeks after primary infection or after a period of asymptomatic TB infection. The bacilli can be found in many tissues, but progression to TB disease in immune hosts occurs primarily in the lungs. The *de novo* formation of secondary TB lesions occurs at the sites distant from the primary lesions, as evidenced by the simultaneous presence of the active TB lesions, often found in lung apices, and calcified inactive primary lesions elsewhere in the lungs [18].

Classic TB treatises [3] and recent studies by Hunter and colleagues emphasize the pathomorphological distinctions between primary and secondary, or post-primary, PTB lesions in human patients [18–21]. The primary TB lesions develop in the lungs after direct aerosol deposition of single Mtb bacteria into intact alveoli. They are represented by organized TB granulomas with characteristic central necrosis surrounded by concentric layers of immune cells and fibrotic tissue. In contrast, the pulmonary TB lesions that develop after the hematogenous Mtb spread morphologically represent areas of granulomatous pneumonia that undergo necrotization leading to massive lung tissue damage with the formation of aerated cavities (reviewed in [15,20,21]). In the immunocompetent hosts, these lesions are walled off by inflammatory tissue from the intact lung and facilitate Mtb transmission by allowing highly infectious individuals remain active despite extensive local lung damage before they are diagnosed or succumb to infection [9,22].

As in humans, lung tissue in mammals is also particularly vulnerable to TB progression irrespective of a route of infection [2,13,14,16]. Summarizing many independent mouse studies, Robert North postulated that the murine model was particularly suitable for the mechanistic dissection of the lung vulnerability [16]. For example, in immunocompetent inbred mice, pulmonary TB develops after systemic intravenous infection. Although only 0.1% of the inoculum is initially distributed to the lungs, within a month after infection the bacterial loads in the lungs are orders of magnitude higher than at the sites of the initial Mtb deposition in the spleen and liver. This lung specificity is lost in immunodeficient *scid* mice, where fulminant TB growth leads to disorganized necrotic inflammation in other organs as well [23]. These observations suggest that Mtb-specific adaptive immunity is necessary and often sufficient for systemic Mtb control, but it is less efficient in the lungs.

A delayed T cell recruitment and/or inadequate access to the sites of infection were proposed to explain failures of T cell-mediated immunity in the lung lesions (reviewed in [4,5,24–26]). Alternatively, the detrimental effect of immune activation, initially described as the Koch phenomenon, is regarded as a plausible mechanism of pulmonary tissue damage impeding the development of safe immunotherapy for TB disease [5,20,27]. Along these lines, it has been proposed that evolutionary conserved Mtb antigens induce T cell-mediated lung damage to facilitate Mtb transmission [28,29].

Among non-immunological mechanisms of the lung vulnerability to TB, the aeration of lung tissue has been considered as a likely mechanism of Mtb lung tropism after hematogenous spread, because in vitro the bacteria replicate faster in

aerated cultures. Several studies also suggest the possible involvement of lung epithelium and fibroblasts and mesenchymal cells in the formation and regulation of PTB lesions [30–33].

Currently, most animal TB models including the mouse model introduce Mtb directly into the lung by aerosol infection or intranasal or intratracheal instillation, thus recapitulating various trajectories of primary TB. Recognizing the need for animal models recapitulating PTB progression after the hematogenous spread, a mouse TB model has been tested where primary TB lesions were induced through intradermal Mtb infection [34]. However, in the common inbred mouse strain genetically resistant to virulent Mtb, C57BL/6J, the intradermal Mtb infection induced T cell-mediated immunity and protection against subsequent aerosol challenge [35]. In this model, pulmonary TB developed only after suppression of essential mechanisms of anti-tuberculosis immunity, such as depletion of NO, neutralization of IFNγ, or CD4 + T cell depletion [36,37].

Here, we describe the progression of human-like post-primary PTB lesions after subcutaneous infection in mice that carry the Mtb susceptibility allele at the *sst1* (*supersusceptibility to tuberculosis* 1) locus [38,39]. Previously, we found that the *sst1* susceptibility allele was specifically responsible for massive lung necrosis after intravenous infection and for the formation of chronic necrotic TB granulomas after a low dose aerosol infection [23,39]. Using reciprocal bone marrow chimeras and in vitro studies, we found that the *sst1*-mediated phenotype was expressed by the bone marrow-derived myeloid cells [23,40] and associated with persistent stress and the hyperactivity of type I interferon in activated macrophages [41–43]. Although these findings might predict widespread infection in multiple organs, the phenotypic expression of the *sst1*-mediated susceptibility in vivo was lung-specific and clearly distinct from acute TB with necrotic lung inflammation observed in immunodeficient *scid* mice [23].

In this study, we characterized PTB lesions that develop after hematogenous spread from remote primary lesions in both the resistant wild type C57BL/6J (B6) and *sst1* susceptible congenic B6.Sst1S mice. While the dissemination and the formation of early interstitial pulmonary microlesions occurred in both strains, only in the *sst1* susceptible mice pulmonary lesions uniquely advanced to necrosuppurative pneumonia that resemble massive lung tissue damage leading to the formation of lung cavities in post-primary PTB in humans [18,19,21,44]. Subsequent mechanistic dissection and experiments using lung implants excluded aeration and revealed important contributions of the aberrant lung epithelial regeneration and alternative macrophage polarization to the formation of a lung-specific microenvironment conducive for PTB progression. We propose that by provoking chronic inflammation and an aberrant lung regeneration in susceptible but immunocompetent hosts, the bacteria gradually create a local niche for its subsequent immune escape, replication and subsequent spread.

## Results

### I. The *sst1* locus drives pulmonary TB progression

To model the hematogenous TB spread in immune animals, we used the subcutaneous (SQ) hock infection, an established alternative to footpad injection [45]. This model resembling post-primary pulmonary TB has been developed and optimized in our laboratory [46]. First, it allows clear separation of the primary site of infection from secondary, metastatic, lesions. Second, the only anatomically possible route of lung colonization is through hematogenous dissemination, resembling post-primary TB progression in humans. Third, the SQ hock injection is a common route of immunization that rapidly primes Mtb-specific T cells in the regional (popliteal) lymph node [35,47]. Comparing the course of infection in the wild type B6 and congenic B6.Sst1S mice, we observed weight loss and decreased survival only in the B6.Sst1S mice. During 20 weeks post infection(wpi), 35% of these animals reached the terminal stage featuring respiratory distress, rapid deterioration of clinical status and severe coalescing granulomatous lesions and necrosuppurative pneumonia (Fig 1A-D). Quantitative analysis of pulmonary inflammation demonstrated an overall increased inflammation and marked heterogeneity of the B6.Sst1S lung lesions at both timepoints, as compared to B6 (Figs 1C and S1A).

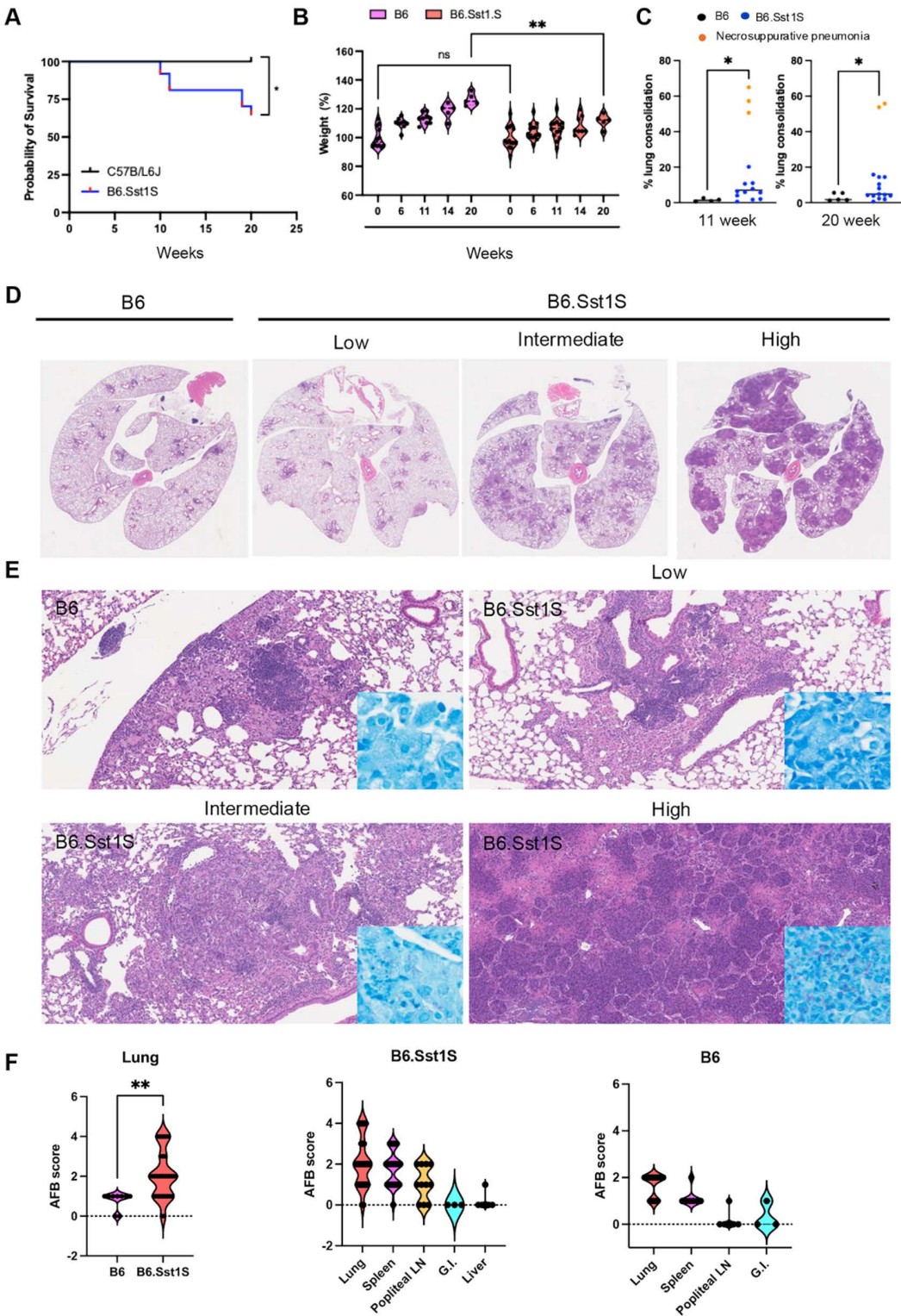

**Fig 1. Comparative Analysis of Survival, Weight Gain, and Pulmonary Pathology in B6 vs. B6.Sst1S Mice Infected with M. tuberculosis.**
A. Survival curves of mice infected with *M. tuberculosis*. Survival of B6 (n = 12) and B6.Sst1S (n = 37) following hock infection with $10^6$ CFU of Mtb Erdman(SSB-GFP, *smyc'*::mCherry). Survival curves were estimated using the Kaplan–Meier method. Differences between groups were assessed using the Mantel-Cox (Log-rank) test. B. Percent weight in mice infected with *M. tuberculosis*. Percent weight of B6 (n = 12) and B6.Sst1S (n = 37) following

hock infection with $10^6$ CFU of Mtb Erdman(SSB-GFP, smyc'::mCherry). The statistical significance was performed by two-way ANOVA using Tukey's multiple comparison test. C. Quantification of lung inflammation in B6 and B6.Sst1S mice infected with Mtb Erdman at 11 (B6, n=4, B6.Sst1S, n=14) and 20 weeks (B6, n=5, B6.Sst1S, n=15) post-infection. Orange dots represent B6.Sst1S mice displaying necrosuppurative pneumonia. D. Representative low-magnification (1X) histopathology images of lung sections from B6 and B6.Sst1S mice following Mtb infection, showing the early disease stage observed in B6 mice and multiple stages of disease progression in B6.Sst1S mice. E. Representative histopathology and AFB staining of lung sections from B6 and B6.Sst1S mice following Mtb infection, illustrating the early disease stage in B6 mice (top left) and various stages of disease progression in B6.Sst1S mice: stage I (top right), stage II (bottom left), and stage III (bottom right). F. Histopathological scores representing *M. tuberculosis* loads in various organs of B6 and B6.Sst1S mice. Each dot represents a single animal. Sample sizes for B6 mice were: lungs (n=12), spleen (n=7), popliteal lymph node (n=6), and gastrointestinal tract (n=3). For B6.Sst1S mice, samples included lungs (n=32), spleen (n=23), popliteal lymph node (n=11), gastrointestinal tract (n=3), and liver (n=6). Statistical significance was assessed using a two-tailed unpaired t-test. The infection experiment comparing B6 and B6.Sst1S mice was performed twice, while B6.Sst1S mice were infected an additional three times, for a total of five infection experiments involving B6.Sst1S. Significant differences are indicated with asterisks (*, P<0.05; **, P<0.01; ***, P<0.001; ****, P<0.0001).

Groups of B6 and surviving B6.Sst1S mice were sacrificed at 11 and 20 wpi. PTB lesions were found in both strains. The B6.Sst1S lesions displayed broad variability, including severe lesions characterized by the presence of cholesterol clefts, neutrophils and areas of necrosis (Fig 1E). In contrast, the B6 lungs had only minimal to mild granulomatous interstitial lesions at both timepoints (S1B Fig).

The bacterial loads were assessed using acid fast staining according to semi-quantitative criteria presented in S1C Fig. PTB lesions in B6 mice, contained rare single acid-fast bacilli-AFB (Fig 1F and S1 Table). The B6.Sst1S PTB lesions were divided into 3 categories with low, intermediate, and high bacterial loads (S2 Table). PTB lesions with low bacterial loads were similar to B6. The intermediate lesions were characterized by increased 1) bacterial loads, 2) the fraction of infected macrophages, 3) the number of intracellular bacilli, and clustering of Mtb bacilli inside infected macrophages. Severe necrosuppurative lesions observed exclusively in B6.Sst1S mice contained innumerable AFB (Figs 1E and S1C), areas of complete tissue necrosis, bronchial occlusion with cell debris containing neutrophils, macrophages and acid fast bacilli, and thrombosed blood vessels (S1D-S1G Fig).

At 6 weeks post-infection, no acid-fast bacilli (AFB) or lesions were detected in the lungs (S2A Fig). However, Mtb was present in the popliteal lymph nodes (S2B Fig). At 12 – 20 wpi, AFB could also be found in the spleen, liver, and gut-associated lymphoid tissue (Fig 1F and and S2 and S3 Tables). In these organs, however, Mtb localized exclusively within small macrophage aggregates with no evidence of extensive bacterial replication, neutrophil influx, or necrosis (S2B-S2E Fig). In mice with high lung Mtb burdens, lung bacterial loads always exceeded those in other organs. In mice with intermediate or low lung burdens, no other organ had higher bacterial loads than the lungs (S2F Fig).

Taken together, these data demonstrate the hematogenous spread of Mtb to many organs after SQ hock infection and PTB progression specifically in B6.Sst1S mice, featuring heterogeneous lesions that range from limited interstitial granulomatous inflammation to bronchogenic spread and lethal necrosuppurative pneumonia. The development of individual PTB lesions in our model occurs 6 – 12 wpi, i.e., after the induction of adaptive immunity. Also it is a relatively rare event – assuming that single Mtb can give rise to a lesion, it is less than 0.0001% of the inoculum that establishes PTB lesions.

## II. Mtb control in the incipient PTB lesions

The incipient microscopic PTB lesions with very low bacterial loads were found in both B6 and B6.Sst1S mice. They developed interstitially in the vicinity of blood vessels and featured prominent lymphoid follicles containing lymphoplasmacytic infiltrates (Fig 2A). To exclude a possibility that lesion formation was driven by non-acid-fast forms of Mtb, we combined acid fast staining with IHC using polyclonal anti-mycobacterial antibodies – both methods produced an overlapping pattern confirming that the incipient lesions contained very few infected macrophages (S3A Fig). To further evaluate this observation, we infected mice with Mtb Erdman expressing constitutive fluorescent reporter mCherry and a replication reporter [48]. To survey a greater portion of the inflammatory lung tissue, we imaged the reporter bacteria in 50 – 100 µm

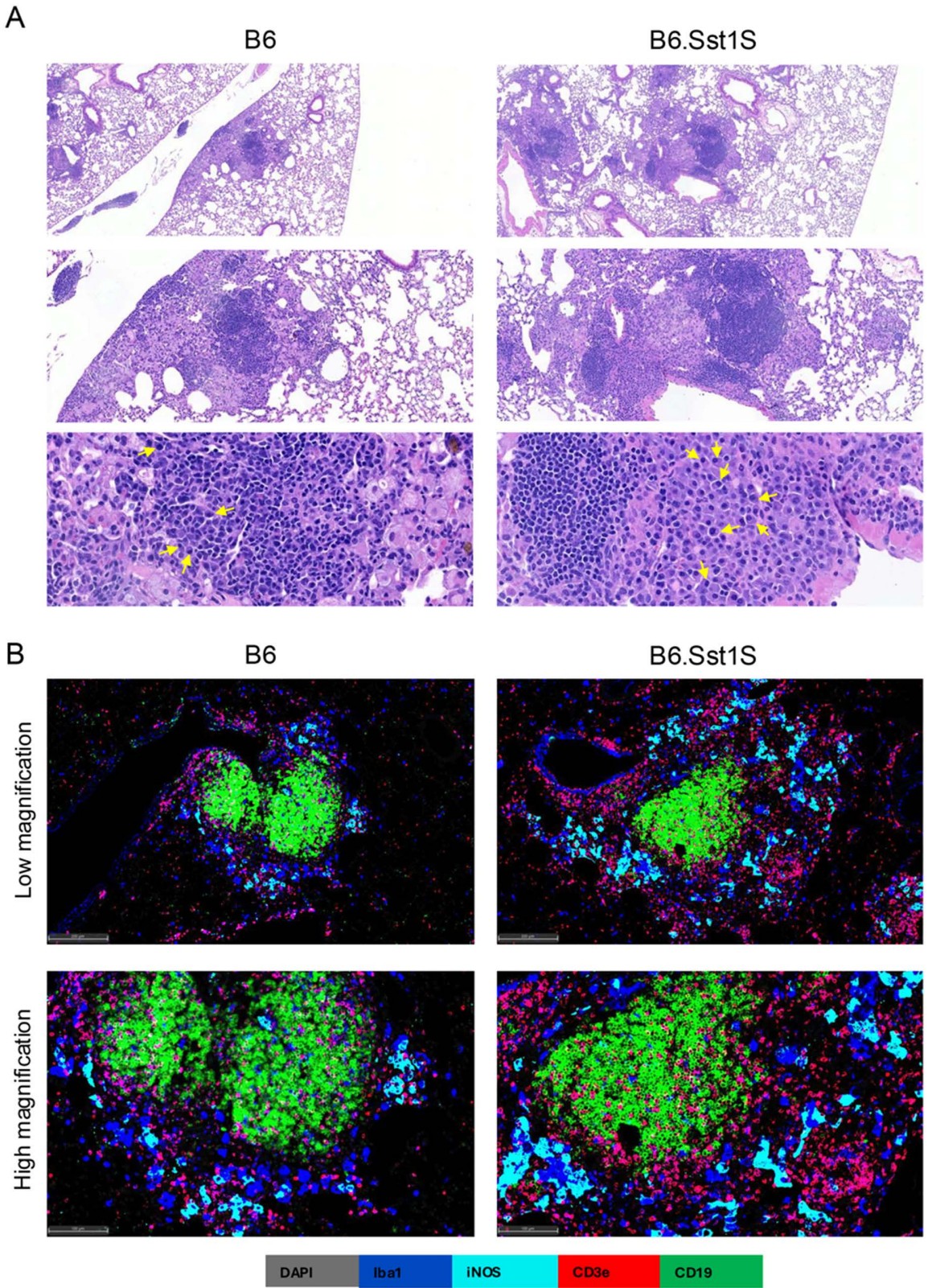

**Fig 2. B cell follicles in early TB lesions.** A. Paucibacillary lesions in B6 and B6.Sst1S mice at 11 weeks post-infection (wpi) (H&E staining, 8X, 200X and 800X original magnification). Plasma cells in intralesional lymphoid follicles are characterized by polarized nuclei, strongly basophilic cytoplasm, and

an eosinophilic perinuclear Golgi apparatus (arrows). B. Fluorescent multiplex immunohistochemistry (fmIHC) of TB lung lesions in B6 (left column) and B6.Sst1S (right column) mice, shown at low and high magnifications, highlights prominent lymphoid follicles.

thick sections using confocal microscopy after tissue clearing. This method also demonstrated that the bacteria in the incipient lesions were mostly represented by single non-replicating bacilli (S3B Fig, left panel).

To further compare the incipient pulmonary lesion composition in B6 and B6.Sst1S mice, we employed fluorescent multiplexed immunohistochemistry (fmIHC) using myeloid markers Iba1 and iNOS, and CD3ε and CD19 for the detection of activated macrophages, and T and B lymphocytes. In both genetic backgrounds the incipient lesions prominently featured organized CD19+B cell follicles and numerous CD3ε+T cells. The myeloid compartment was represented by Iba1+macrophages, a fraction of which was activated as evidenced by iNOS+ expression (Fig 2B). Sparse individual bacteria localized exclusively inside macrophages and the vast majority of iNOS+ activated macrophages contained no acid-fast bacilli.

These data demonstrate that after hematogenous spread the incipient TB lesions emerge in lung interstitium in the vicinity of blood vessels and feature active adaptive and innate immune responses. This response is sufficient for the initial immune-mediated control of Mtb in both genetic backgrounds. As compared to other organs, however, the local response seems to be disproportionate to the bacterial loads and precedes the bacterial growth.

### III. PTB progression in B6.Sst1S mice

Next, we compared PTB lesions of the *sst1*-susceptible mice that either controlled or sustained Mtb replication - paucibacillary (P-type) and multibacillary (M-type), respectively. Of note, advanced necrotizing lesions (Fig 1E, lower right panel, (S1C (4+), S1D-S1E and S3B Figs, right panel) were excluded from this analysis. Compared to P lesions described above, the M lesions contained replicating bacilli, increased proportion of iNOS+ activated macrophages, and areas of micronecrosis with neutrophil infiltration (Figs 3A and 3B and S3B). The M lesions did not contain organized lymphoid follicles (Figs 3C and S3C), although the overall B and T cell content per lesion did not decline (Fig 3D). Thus, the failure to control Mtb in PTB lesions of the susceptible hosts was associated with disorganization of lymphoid tissue and expansion of myeloid cell populations.

To identify the underlying mechanisms, we used the Nanostring GeoMx Digital Spatial Profiler (DSP) spatial transcriptomic platform to compare representative P and M lesions that developed in the susceptible mice at 14 wpi. To focus on monocyte/macrophage population within pulmonary TB lesions, we selected macrophage-rich areas as regions of interest (ROI) avoiding areas of necrosis and tertiary lymphoid tissue, with uninvolved (U) lung areas serving as controls. A total of eight areas of each (uninvolved lung, P and M lesions) were selected and analyzed using the Mouse Whole Transcriptome Atlas (WTA) panel which targets 21,000+transcripts for mouse protein coding genes. The analysis identified 793 differentially expressed genes (DEG) with q≤0.05 (S4 Table). Both Principal Component Analyses (PCA) and hierarchical clustering demonstrated remarkable concordance within the groups and consistent differences between the groups (Fig 3E).

For digital deconvolution of the cellular content of the ROIs we used CIBERSORTx software (Fig 3F). Consistent with fmIHC, this analysis demonstrated decreased proportions of T, B, plasma cells and monocytes, and increased proportions of macrophages and neutrophils in M-type lesions.

Using the DEG sets, we performed Gene Set Enrichment Analysis (GSEA) [49]. The top 10 pathways upregulated in M lesions are listed in S5 Table. Among them, the upregulation of genes in inflammatory pathways, TNF and IL-6 signaling, and complement pathways were consistent with increased macrophage activation. An increased hypoxia pathway activation was consistent with increased size of the M lesions and intravascular thrombosis observed on tissue sections. Confirming the escalating hypoxia, the accumulation and nuclear translocation of HIF1α significantly increased within the M lesions (Fig 3G and 3H).

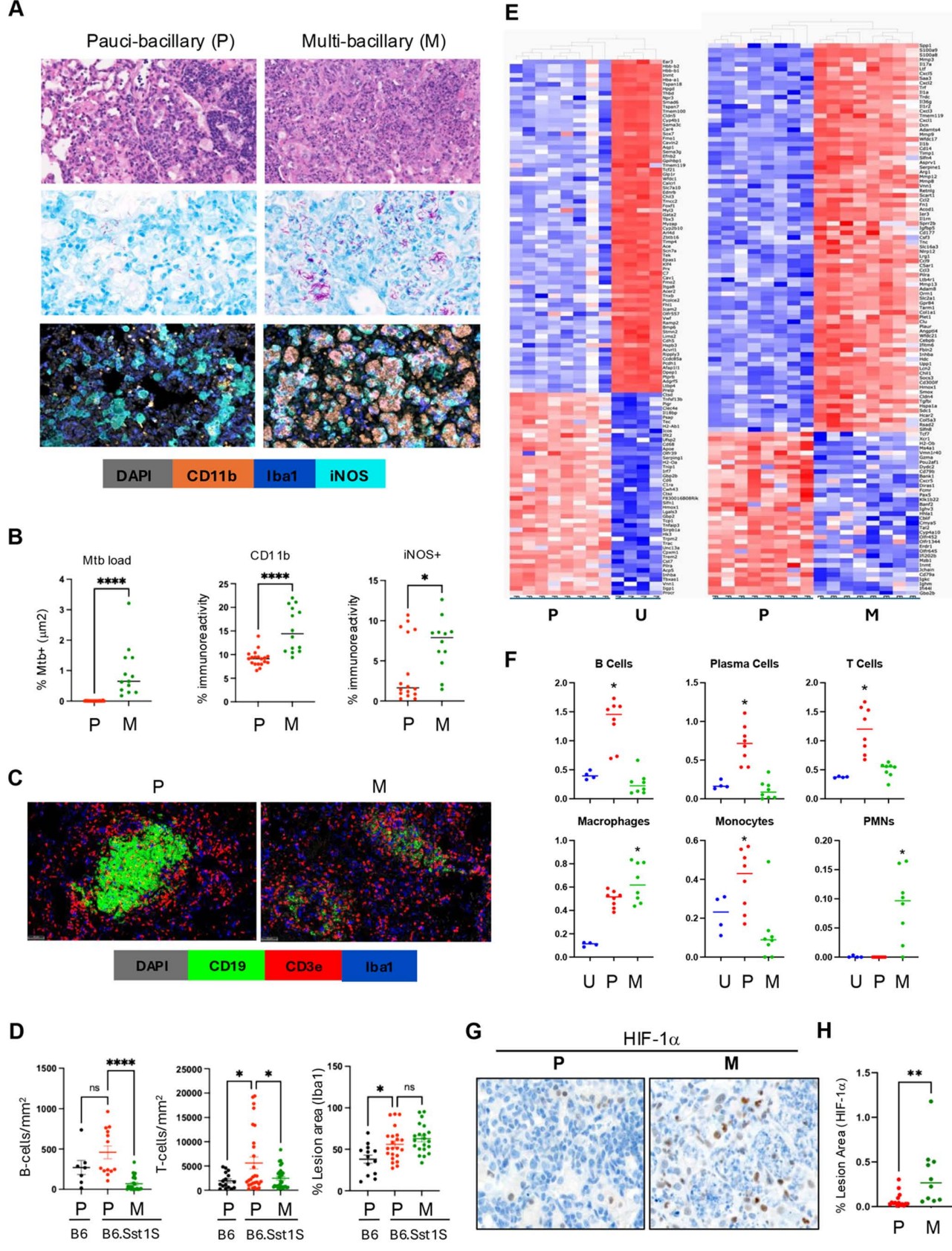

**Fig 3. Comparative Cellular and Spatial Transcriptomic Analysis of Paucibacillary and Multibacillary pulmonary TB Lesions of B6.Sst1S Mice.** A. Representative images of paucibacillary and multibacillary lesions from B6.Sst1S lung: H&E (top row), acid-fast bacilli (AFB) (middle row) and CD11b immunohistochemistry (bottom row) at 14wpi. The M lesions had areas of micronecrosis, neutrophilic influx, and AFB. B. Quantification of Mtb load and myeloid cell populations in paucibacillary (P) versus multibacillary (M) lesions using the HALO area quantification algorithm. Lung sections of a total of eight B6.Sst1S mice were analyzed. The number of lesions analyzed: for Mtb load P (n = 20) and M (n = 13); for CD11b+ cells, P (n = 19) and M (n = 14); and for iNOS+ cells, P (n = 16) and M (n = 12). Statistical significance was determined by two-tailed unpaired t-test. C. Representative fluorescent multiplex immunohistochemistry (fmIHC) of paucibacillary (left) and multi-bacillary (right) lesions. Ionized calcium-binding adaptor molecule 1 (Iba1) - blue; CD19 - green; CD3 epsilon (CD3ε) – red and DAPI - grey. 200x total magnification. D. Comparisons of pauci- and multibacillary pulmonary TB lesions of B6 and B6.Sst1S mice. B cell and T cell densities, and Iba1+cell area were quantified using HALO algorithm. A total of 5 mice per group were analyzed. Total number of B6 paucibacillary lesions analyzed: for: B cells (n = 7), T cells (n = 16), and Iba1+cells (n = 13); for B6.Sst1S paucibacillary lesions: B cells (n = 13), T cells (n = 28), and Iba1+cells (n = 22); for B6.Sst1S multibacillary lesions: B cells (n = 24), T cells (n = 31), and Iba1+cells (n = 21). Statistical significance was determined using ordinary one-way ANOVA with Bonferroni's multiple comparison test. E. Heatmaps of GeoMx spatial transcriptomic analysis comparing uninvolved lung (U, n = 4), paucibacillary (P, n = 8) and multibacillary (M, n = 8) lung lesions from B6.Sst1S mice. Lung lobes were selected from 2 mice with paucibacilarly lesions and 2 mice with multibacillary lesions with micronecrotic areas. The four lobes were assembled on one slide and processed in parallel. Slides were stained with CD45-, pan-keratin-specific antibodies and DAPI. Gene expression, Red-High and Blue-Low. F. Cellular composition of pulmonary TB lesions analyzed using GeoMx spatial transcriptomics (panel 3E) were deconvoluted using CYBERsort algorithm. Statistical analysis was performed using Kruskal-Wallis nonparametric ANOVA for multiple comparisons. G and H. Nuclear HIF1α staining of paucibacillary (P) vs multibacillary (M) lesions. Representative immunohistochemistry images (G) and percent lesion area quantification by HALO algorithm (H) of hypoxia-inducible factor 1-alpha (HIF-1α) expression in P (n = 12) and M (n = 10) TB lesions. 200x total magnification. The statistical significance was performed by two-tailed unpaired t test. The p value <0.05 was considered statistically significant. Significant differences are indicated with asterisks (*, P < 0.05; **, P < 0.01; ***, P < 0.001; ****, P < 0.0001).

Consistent with histopathology (Fig 3A), we observed the upregulation of neutrophil-related genes, such as neutrophil proteins (*S100a8 and S100a9, CD177, Lcn2, Lrg1*), chemoattractants (*Cxcl1, Cxcl2, Cxcl3, Cxcl5, IL-17, IL36g*), and neutrophil growth factor (*CSF3/G-CSF*). Also, we noted the upregulation of type I interferon (IFN-I) stimulated genes (*Acod1, Rsad2, Il1rn, Ifitm6, Slnf4, Slfn8*), metalloproteases (*Mmp3, Mmp8, Mmp9, Mmp12, Mmp13, Adam8, Adamts4*), and markers of oxidative (*Hmox1*) and proteotoxic (*Hspa1a*) stress.

In parallel, we observed a coordinated upregulation of markers of M2 macrophage polarization (*Arg1, Arg2, Fizz1/resistin-like beta/gamma, Chil1, CD300lf*). This finding was surprising, because M lesions also contained increased proportions of M1-polarized iNOS+ macrophages (Fig 3A and 3B). However, fmIHC confirmed the appearance of Arg1 + macrophages (Iba1+) in M lesions (Fig 4A) concomitantly with the increased proportion of iNOS+ macrophages (Fig 4B). Of note, the Arg1+ and iNOS+ macrophage populations were clearly distinct, as shown by spatial analysis using HALO (Fig 4B). Interestingly, these macrophage populations were often juxtaposed, resembling cloaking of the iNOS+ macrophages by the Arg1 + macrophages and possibly sequestering them from T lymphocytes (Fig 4C).

Thus, spatial transcriptomic and fmIHC analyses revealed that the loss of Mtb control in susceptible hosts was associated with a coordinated transition from lymphocyte-dominated to myeloid-dominated inflammatory lesions characterized by the increased proportion of iNOS+ activated macrophages and a concomitant appearance of a distinct population of alternatively activated Arg1 + macrophages.

## IV. Exploring the origin of the Arg1 + macrophages

First, we sought to determine whether the Arg1 + cells expand locally via proliferation. Using co-staining of M lesions with the proliferation marker Ki67, we found that most of the proliferating macrophages in M lesions were Arg1-negative and iNOS-positive (Fig 4D-E). Thus, local proliferation does not explain the Arg1 + macrophage expansion. Next, we co-stained Arg1 + cells with known markers of lung-resident macrophages (CD206 and CD163) and recruited myeloid cells (CD11b). Comparing lungs of non-infected mice, uninvolved areas of Mtb-infected lungs and M-type PTB lesions, we found that Arg1 is expressed by a fraction of the lung resident (CD206+ or CD163+) macrophages in uninfected areas. However, the absolute majority of Arg1 + cells within the M lesions were CD11b-positive (Fig 4F-G). These data indicate that the Arg1 + cells in M lung lesions primarily originate from recruited inflammatory monocytes.

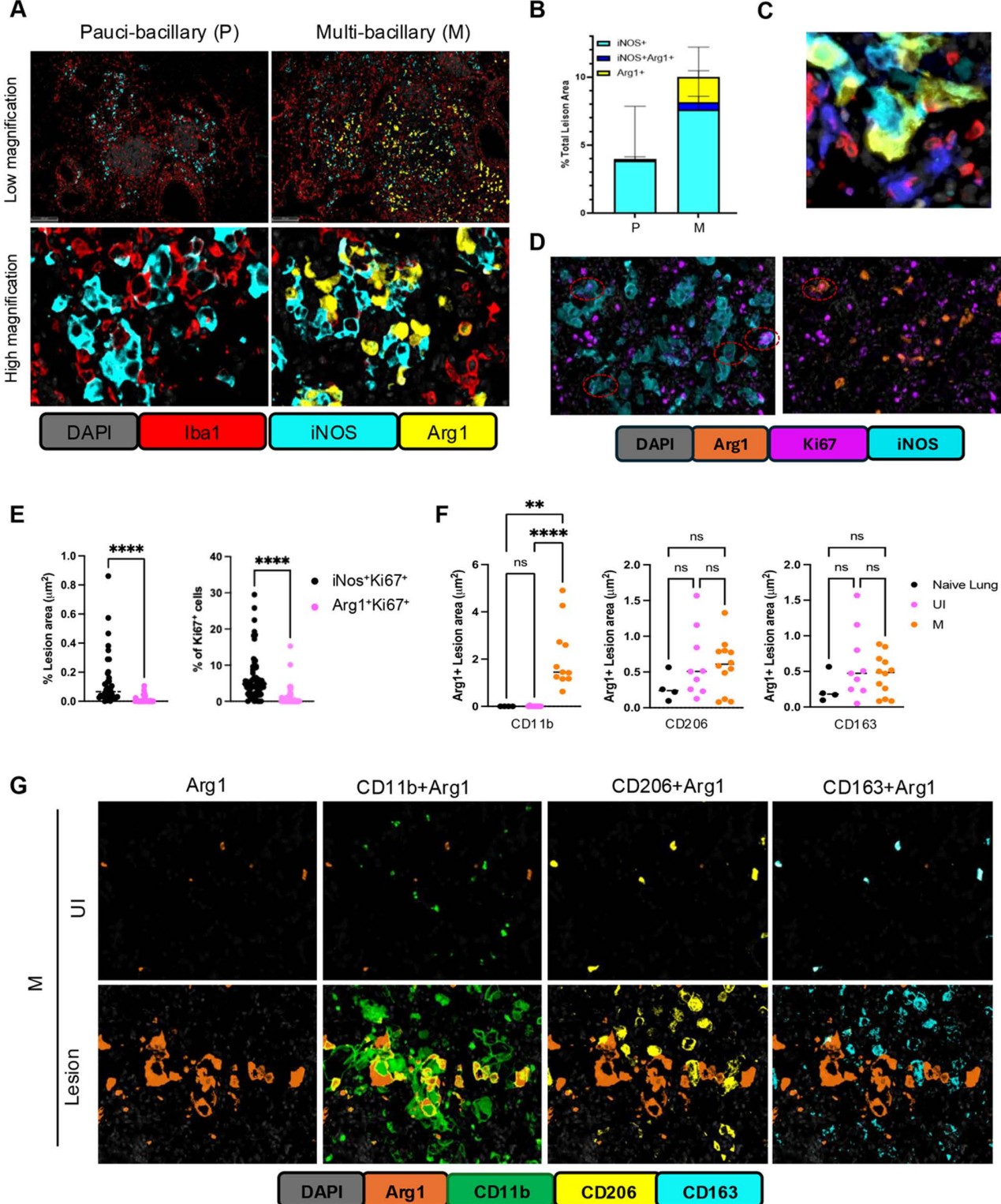

**Fig 4. Characterization of Arg1+cell in pulmonary TB lesions of B6.Sst1S mice.** A. Representative fluorescent multiplex immunohistochemistry (fmIHC) of paucibacillary (left) and multi-bacillary (right) lesions at low and high magnifications. Arginase 1 (Arg1) - yellow; inducible nitric oxide (iNOS) - teal; ionized calcium-binding adaptor molecule 1 (Iba1) - red and DAPI - grey. B. Quantitative analysis of iNOS+Arg1-, iNOS-Arg1+ and

iNOS+Arg1+ cells in P (n = 5) and M (n = 5) lung lesions of B6.Sst1S mice. C. High magnification of interacting Arg1 - (yellow) and iNOS (teal) – expressing macrophages. T cells – red, Iba1+iNOS-Arg1- macrophages – blue. D. FmIHC images showing proliferation of Arg1+ (Top panel) and iNOS+ (lower panel) cells (marked as circle) in TB lesion area. The proliferation was observed and quantified using staining of tissues with Ki67 markers. E. Quantification of Ki67+ macrophage populations. Left panel – area quantification of iNos+Ki67+ and Arg1+K67+ cells within multibacillary PTB lesions (n = 38). Right panel - percentage of Ki67+ cells that are either iNOS+ or Arg1+ within individual lesions (n = 60). Total 6 mice were analyzed. Statistical significance was determined using the two-tailed Mann–Whitney test. F and G. Quantification of Arg1+ macrophages co-expressing CD11b, CD206, or CD163, representing recruited and resident myeloid cell populations in naïve lung (n = 4), Mtb infected lung uninvolved area (n = 9) and multibacillary PTB lesion area (n = 12). (F). Representative fmIHC images showing Arg1+, Arg1+CD11b+, Arg1+CD206+ and Arg1+CD163+ populations (G). In the images, CD11b (green) marks recruited myeloid cells, while CD206 (yellow) and CD163 (teal) indicate resident myeloid cells. Arg1+ cells are shown in orange. A total of 5 mice were analyzed from Mtb infected groups. The statistical significance was performed by ordinary one-way Anova using Sidak's multiple comparison test. The p value <0.05 was considered statistically significant. Significant differences are indicated with asterisks (*, P < 0.05; **, P < 0.01; ***, P < 0.001; ****, P < 0.0001).

To determine whether recruited cells acquire the alternative activation phenotype locally or prior to their recruitment to the lesions, we performed blood transcriptomics analysis. Total RNA was isolated from the blood obtained from uninfected and Mtb-infected mice. The latter were sacrificed and their PTB lesions were classified as paucibacillary or multibacillary using histopathology and acid-fast staining. A DriverMap mouse gene expression panel was used to analyze the expression of ~4,800 genes. Comparison between infected and non-infected controls identified 87 upregulated and 90 downregulated genes that reached a significance level of p < 0.05 and exhibited a log2-fold change. The upregulation of *Irgm2, Gbp2, Gbp4, Gbp5, Gdf3*, and *Maf* genes reliably differentiated between non-infected and Mtb-infected blood samples (Figs 5A, 5B, S4A and S4B).

We also noted the upregulation of M2 markers *Arg1* and *Chil3* mRNA in the blood of mice with advanced PTB lesions and used qRT-PCR to validate the differential gene expression in independent blood samples isolated from naïve and Mtb-infected mice. Significant upregulation of *Arg1* and *Chil3* mRNAs was found in mice with paucibacillary lesions, reaching its highest levels in animals with multibacillary lesions (Fig 5C). The *Il6* mRNA expression was modestly elevated only in mice with multibacillary lesions. No upregulation of Ifnβ gene (*Ifnb1), Rsad2, Trib3* and *Chac1* mRNAs was detected in the blood of Mtb-infected mice (Figs 5C and S4C). These data demonstrate the dissociation between macrophage polarization and the aberrant activation and stress reported previously [41].

To further investigate macrophage polarization signatures in the blood of Mtb-infected animals, we used the MacSpectrum software that was developed to map macrophage activation states under both polarizing and non-polarizing conditions using M1 and M2 gene signatures [50]. The M1 and M2 polarization gene sets were partially upregulated in mice with both paucibacillary and multibacillary PTB lesions, as compared to uninfected controls (S4D-S4E Fig), as was also demonstrated by the increase of the M1 and M2 macrophage polarization indices calculated using MacSpectrum (Fig 5D). The M2 polarization index was more prominent in the blood transcriptome of mice with multibacillary PTB lesions, as compared to mice with the paucibacillary lesions (Fig 5D-E).

To investigate whether the Arg1+ cells in circulation are of bone marrow origin, we used spectral flowcytometry with an extended antibody panel (S6 Table and S5A Fig). Upon infection, we observed > 4-fold increase in the CD45+Arg1+ cells (S5B Fig and S7 Table). Most of the CD45+Arg1+ cells belonged to two subpopulations: CD11b+Ly6C+ monocytes and CD11b-Ly6C-Sca1+ cells (Fig 5F-G). These results indicate that Arg1 expression is initiated in Sca1+ progenitors in bone marrow and Mtb-infection induces the expansion of Arg1+CD11b+Ly6C+ monocytes committed to M2-like differentiation prior to their recruitment to PTB lesions.

## V. Lung-specific TB progression

In our spatial transcriptomics data Epithelial-Mesenchimal Transition (EMT) and K-ras signaling were among the top 10 pathways upregulated in the advanced lesions, suggesting perturbation of lung epithelial cells within the TB lesions (S5 Table). Staining of PTB lesions with antibodies recognizing lung epithelial marker NK2 homeobox 1 (Nkx2.1/ Ttf1),

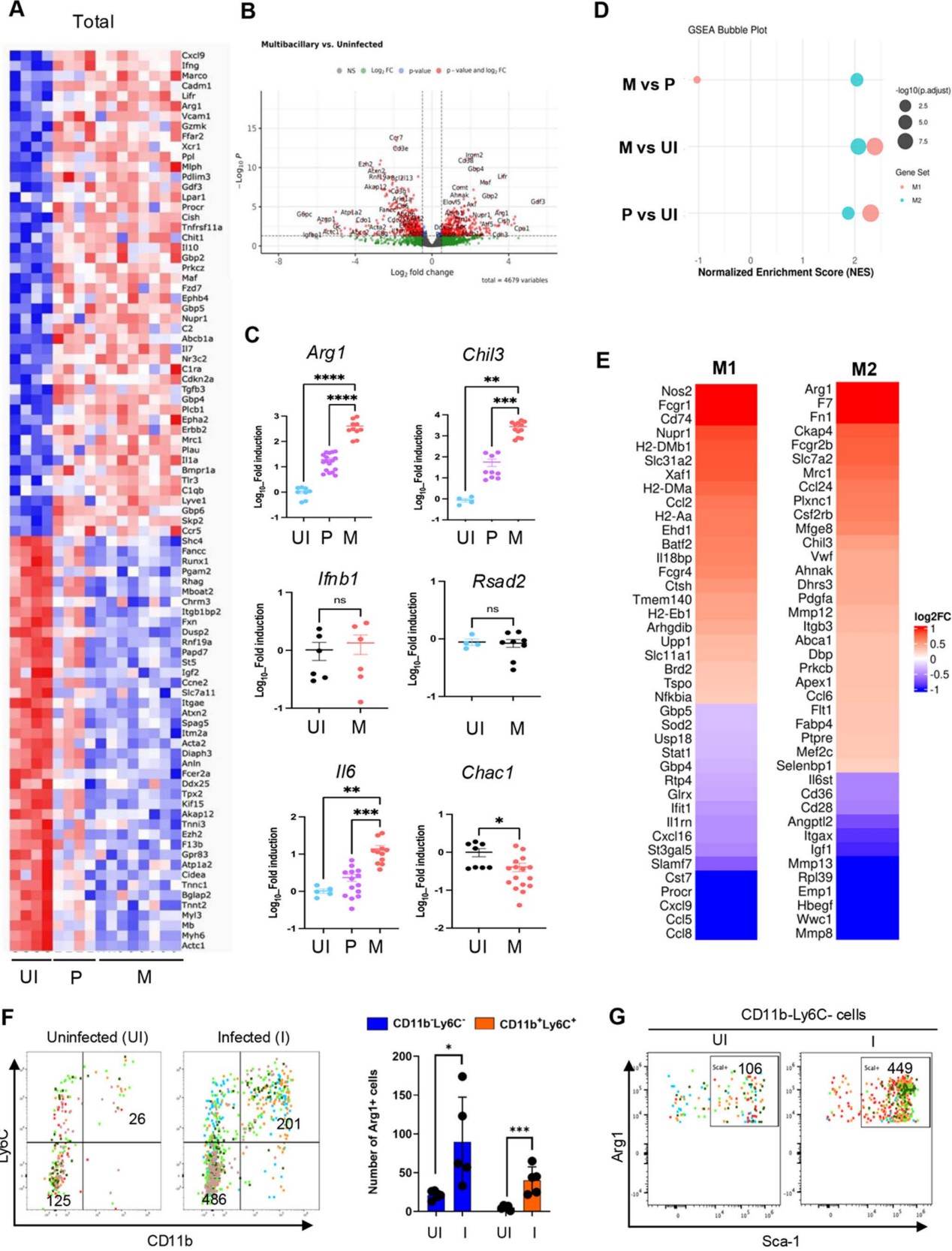

**Fig 5. M2 polarization of circulating myeloid cells.** A. Heatmaps depicting the differentially expressed genes (DEGs) in the blood of mice, categorized as uninfected controls or mice with either P or M lung lesions. Gene expression levels were quantified by blood transcriptomics, highlighting the distinct transcriptional profiles between these groups. B. Volcano plots showing differentially expressed genes in blood samples from *M. tuberculosis*-infected mice with multibacillary lung lesions compared to non-infected controls. Each point represents a single gene, with the x-axis indicating log2 fold change and the y-axis showing -log10(p-value). Significantly downregulated and upregulated genes are highlighted in red (left and right, respectively), while non-significant genes are shown in gray. Significance thresholds are set at $p < 0.05$ and fold change $> \pm 1.5$. C. Expression levels of *Arg1, Chil3, Ifnb1, Rsad2, Il6*, and *Chac1* mRNA in the blood of Mtb infected mice with paucibacillary and multibacillary lung lesions, normalized to non-infected control mice. The expression of *Arg1, Chil3* and *Il6* is shown for both paucibacillary and multibacillary lung lesions, while the expression of the other genes are presented for mice with multibacillary lung lesions. Each dot represents a single animal. The statistical significance was performed by two-tailed unpaired t test. D. Blood transcriptome polarization indices calculated using MacSpecrum for comparisons of mice with paucibacillary (P), multibacillary (M) PTB lesions and uninfected control mice (UI). E. Heatmaps showing the expression levels of M1 and M2 macrophage-specific genes in blood samples from mice with multibacillary lesions compared to paucibacillary. F and G. Expansion of Arg1 expressing progenitor cells in the bone marrow during Mtb infection. After the infection, the bone marrow cells were collected and analyzed by flowcytometry for Arg1 expression. F. The dot plot of CD45+Arg1+cells for CD11b and Ly6C expression (Left panel). The bar graph showing the increase in subsets of Arg1 expressing CD11b+Ly6C+ and CD11b-Ly6C- cells upon Mtb infection (Right panel). G. The CD11b-Ly6- cells were further analyzed for Sca1 expression. The dot plot showing that CD11b-Ly6- cells express Sca1. The dot plots of five individual mouse per group were merged into one dot plot. The total numbers of cells from five mice of uninfected or infected groups were shown in the gates. The dots in the bar graph indicate individual animals. The statistical significance was performed by two-way ANOVA using Tukey's multiple comparison test. The p value <0.05 was considered statistically significant. Significant differences are indicated with asterisks (*, $P < 0.05$; **, $P < 0.01$; ***, $P < 0.001$; ****, $P < 0.0001$).

alveolar type I pneumocyte (AT1) marker Caveolin 1 (Cav1), and alveolar type II pneumocyte (AT2) marker pro-surfactant protein C (SPC) revealed a significant proportion of these cells within PTB lesions (S6 Fig). The AT1 cells (Cav1+) embedded within the lesions formed a pattern clearly recapitulating the shape of the alveolar network. Also, PTB lesions contained clusters of nuclear Nkx2.1+cells that were either prosurfactant C (SPC)-positive or negative. These intralesional epithelial cell clusters contained dysplastic regenerative pneumocytes (p63+Krt17+double positive cells and p63- Krt8$^{HI}$ transitional cells (Fig 6A). These data demonstrate that lung epithelium is intricately associated with PTB lesions and display aberrant repair phenotypes.

Recently a subpopulation of lung progenitors induced by chemical injury was found to express an IFN-I signature [51]. Given the importance of the IFN-I pathway in TB susceptibility in mice and human patients [52], we used a B6.Sst1S, *Ifnb1*-YFP reporter mouse strain to determine whether lung epithelial Nkx2.1+cells express IFNβ within TB lesions. To detect the YFP-expressing epithelial cells in PTB lesions of the Mtb-infected B6.Sst1S,*Ifnb1*-YFP mice, we used 3D imaging of cleared thick sections co-stained with Nkx2.1-specific antibodies, as described in [53]. The Nkx2.1+YFP+ double positive cells were found in association with residual alveolar septae (Fig 6B). Another YFP-positive cell population was iNOS+ activated macrophages (Fig 6C). The specificity of YFP signal in Nkx2.1+cells was confirmed using staining of regular paraffin-embedded sections with GFP-specific antibodies (S7A Fig). An artefact due to autofluorescence was excluded by the lack of the YFP signal in PTB lesions of *sst1*-susceptible mice that did not carry the YFP reporter (S7B Fig, right panel). The YFP expression by alveolar and activated fibroblasts was excluded using co-staining with fibroblast markers Col1a1 and smooth muscle actin (SMA) (S7C-S7D Fig, respectively). Thus, lung epithelial cells survive chronic injury within TB lesions, undergo an aberrant regeneration, express IFNβand possibly other immune mediators. Their activities within PTB lesions may uniquely pre-condition lung microenvironment for TB progression.

To test this hypothesis, we implanted fragments of mouse lung tissues subcutaneously into syngeneic B6.Sst1S recipients, as described in detail in [46]. Spleen fragments were also implanted next to the lung. After two months, the recipient mice were subcutaneously infected with Mtb. At the time of sacrifice, we confirmed the implants' viability by staining with Nkx2.1- and Cav-1-specific antibodies (Fig 6D). Mice with advanced TB lesions in native lungs also developed necrotic TB granulomas within the subcutaneously implanted lung tissue fragments. Granulomas within lung implants consisted of organized cell and fibrotic layers surrounding necrotic cores containing clusters of replicating bacteria (Figs 6E and S8A). In contrast, the adjacent spleen implants had no granulomatous lesions and rare acid fast bacilli were found intracellularly within small macrophage clusters (S8B Fig). Abundant Arg1+macrophages were found in multibacillary advanced TB

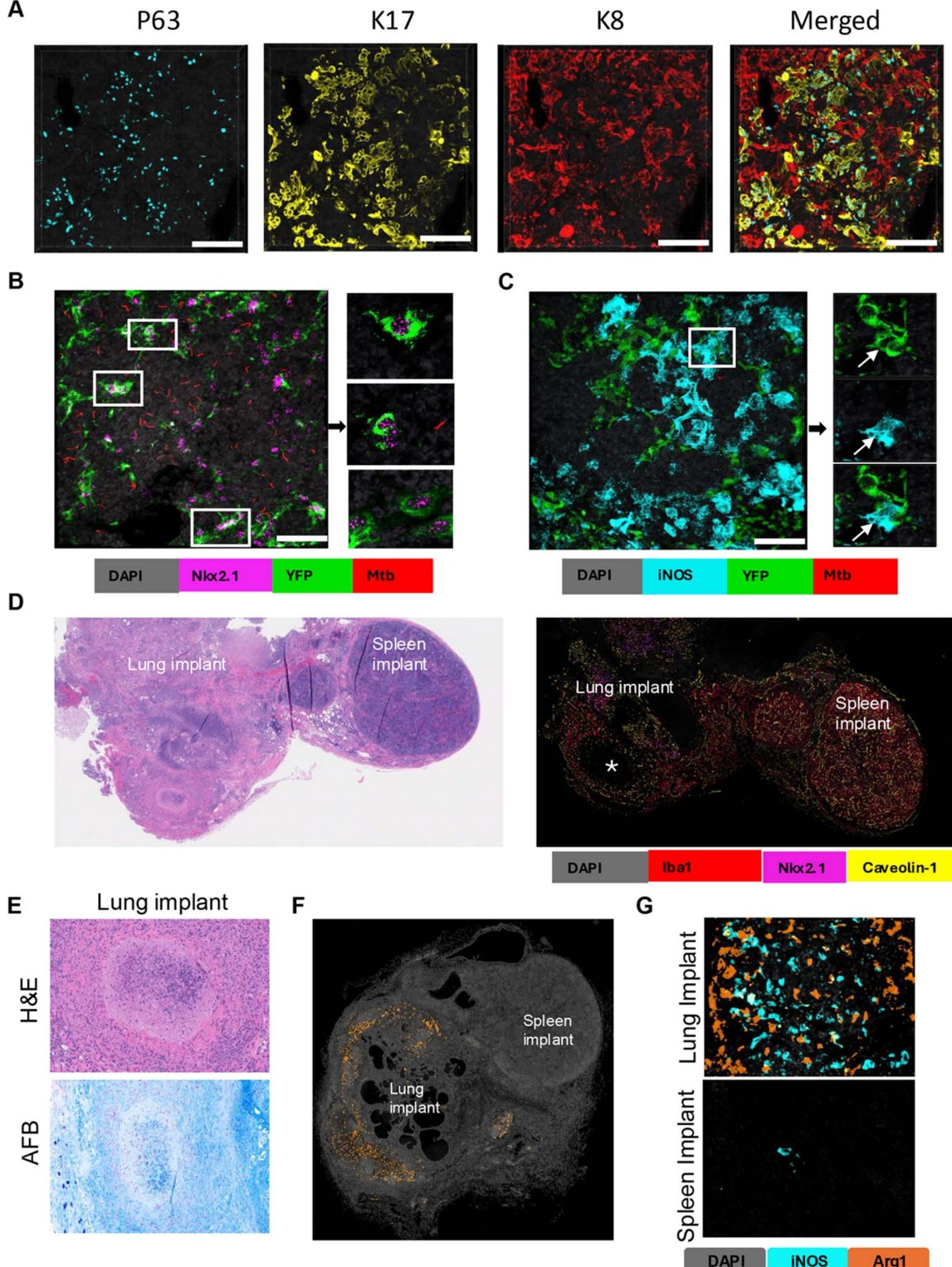

**Fig 6. Mechanisms of lung-specific TB progression.** A. 3D confocal fluorescent multiplex immunohistochemistry (fmIHC) images of 50 µm thick lung sections stained with p63 (teal) Krt17 (yellow), Krt8 (red). Scale bar:100 µm. B. 3D confocal images of cleared thick lung sections (50 µm) of the TB lesion of B6.Sst1S,*Ifnb1*-YFP mice. Endogenous YFP is shown in green, Nkx2.1 staining - in magenta, and reporter Mtb (*smyc'*:: mCherry) in red. Left image - all the stacks in 3D, on the right – individual images of Nkx2.1+cells expressing YFP (denoted by white boxes on the left panel). Scale

bar- 100 μm. C. 3D confocal images of cleared thick lung sections (50 μm) of the TB lesion of B6.Sst1S,*Ifnb1*-YFP mice. Endogenous YFP is shown in green, iNOS staining - in teal, and reporter Mtb (*smyc'*:: mCherry) in red. Left image - all the stacks in 3D, on the right – individual images of iNOS+ cells expressing YFP (denoted by white box on the left panel). Scale bar- 50 μm.D. Adjacent lung (left) and spleen (right) implants under the mouse skin. Left image - H&E. Original magnification - 100X. Right image – fmIHC revealing lung epithelial cells (Nkx2.1+) exclusively in lung implants and Iba1+macrophages in both spleen and lung implants. Organized necrotic granuloma in the lung implant is marked by star.E. Representative hematoxylin and eosin (H&E) and acid-fast bacilli (AFB) staining of lung implant. Lung implant with a focal caseating granuloma surrounded by collagen (top) with innumerable AFB (bottom). 200x total magnification. A total of 10 implants were processed and analyzed. F and G. fmIHC image showing presence of Arg1+ and iNOS+ cells in lung implant but not in spleen implant.

lesions within both native lungs and lung implants and were absent or rarely found in the adjacent spleen implants (Figs 6F-G and S8C). Lung implants of B6 mice in B6.Sst1S recipients also developed necrotic lesions with numerous bacilli demonstrating the *sst1*-independent effect of lung parenchyma on TB progression(S8D Fig).

Expanding our observations, human immune system (HIS) mice carrying subcutaneous implants of human lung tissue were infected with Mtb via respiratory route. Necrotic TB lesions with abundant Mtb were found in mouse lungs (Fig 7A) and subcutaneous human lung implants (Fig 7B) following pulmonary challenge. Taken together, the mouse and human implant models demonstrate the exceptional vulnerability of lung tissue to the hematogenously spread virulent Mtb, and provide novel mechanistic insights and small animal models for further investigation.

## Discussion

Our results clearly demonstrate that pulmonary TB does not require direct deposition of Mtb in the lungs via aerosols and develops after hematogenous spread even in the presence of systemic immunity. The paucity and heterogeneity of lung lesions in our model demonstrate that seeding the lung after hock infection and hematogenous spread are asynchronous and rare events, perhaps not exceeding the seeding frequency in other organs. However, the structure and the dynamics of the pulmonary lesions are drastically different from other organs.

Although individual lung samples for these studies were obtained at animal sacrifice and do not represent a true longitudinal time course experiment, lesion classification according to pathomorphological characteristics and bacterial loads suggested stepwise progression and allowed us to make some stage-specific mechanistic inferences. First, in contrast to primary TB after aerosol infection, a delay of local adaptive immunity does not account for the lung-specific TB progression after hematogenous dissemination. Instead, the paucibacillary lung lesions that developed after several weeks post primary hock infection were the sites of active immune responses represented by tertiary lymphoid tissue containing active B cell follicles with plasma cells, numerous T cells and activated macrophages expressing iNOS, which is necessary for the initial control of Mtb [36] and had a very low bacterial burden. In the resistant B6 mice these lesions were maintained for at least 20 weeks p.i., protecting the host from PTB progression, albeit they were insufficient to eradicate the bacteria. These stable lesions may represent an example of the "arrested" microscopic lesions that control the infection without eradication, as postulated in classical TB studies [2,3,14].

The more advanced multibacillary PTB lesions found in the *sst1* mice were characterized by disorganized tertiary lymphoid tissue and moderate granulomatous pneumonia occupying airspaces. These lesions also contained abundant iNOS+ macrophages, the bacterial loads remained low, and no necrosis and neutrophil infiltration were found. Therefore, we surmise that these lesions are shaped by inflammatory mediators associated with monocyte/macrophage lineages and, possibly, secreted bacterial products, but not an active bacterial replication or neutrophil recruitment. At the advanced and terminal stages, we observed coalescing necrotizing pneumonia with airspaces and airways filled with innumerable neutrophils, abundant necrotic cellular debris, and profound replication of Mtb.

The hyperinflammatory lesion progression is consistent with intrinsic dysregulation of the *sst1* susceptible macrophage activation that we described previously [41,54]. In vitro studies demonstrated an aberrant response of the *sst1*-susceptible macrophages to TNF, a cytokine essential for local TB resistance and granuloma maintenance (reviewed in [55]). It

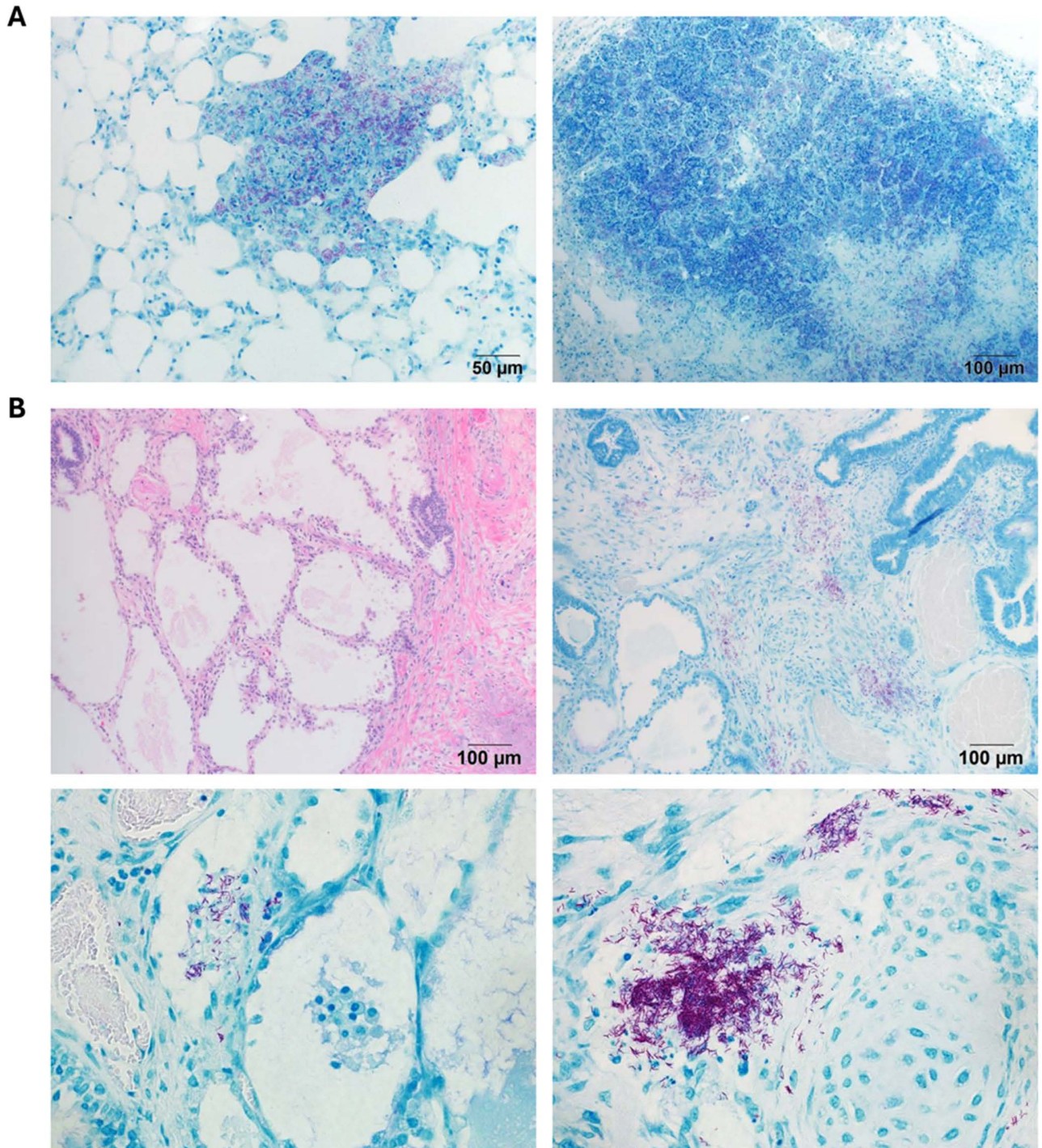

**Fig 7. Mtb traffics to and proliferates in human lung xenografts.** Human lung tissue was engrafted by suturing fragments into the muscle fascia in the dorsal subcutaneous space. A. Representative AFB of lung in HIS mouse demonstrating abundant bacilli in a solid, developing lesion (left, top) and a necrotic granuloma (right, top). B. Human lung explants demonstrating Mtb bacilli and areas of proliferation in human lung xenografts. Representative H&E showcasing lung explant characteristics (left, top) and areas of inflammation with abundant AFB (right, top). High power images (bottom) demonstrating proliferation in inflammatory areas adjacent to blood vessels and alveoli (left, bottom) and inflammatory foci in parenchyma (right, bottom). Bottom panels, 40X.

was characterized by unresolving oxidative and proteotoxic stress and a positive feedback regulation of the stress- and IFN-I-mediated pathways. Recently, the *sst1*-encoded SP140 protein was shown to play a major role in *sst1*-susceptibility in vivo [43]. In human GWAS studies, the SP140 polymorphisms were associated with non-infectious inflammatory diseases - multiple sclerosis and Crohn's disease (reviewed in [56]). Mechanistically, it was linked to re-activation of heterochromatin and the upregulation of developmentally silenced genes, including homeobox genes, and higher overall transcriptional activity including inflammatory genes [57]. Taken together these studies demonstrated that the *sst1* locus controls the inflammatory response, but not the effectors of antimycobacterial immunity. In this context, Mtb serve as a trigger, but the subsequent phenotype amplification is driven by myeloid cells in a cell autonomous manner via secretion of inflammatory mediators, such as IFNβ and the IFN-I-mediated chemokines, and myeloid cell recruitment. However, it remained unknown why this cell-autonomous macrophage mechanism compromised Mtb control specifically in the lungs, despite pre-existing systemic immunity.

This study focuses on the lung-specific aspect of the *sst1*-susceptible phenotype in vivo. Contrary to a common perception of TB lesions as granulomas primarily composed of immune and inflammatory cells, our data demonstrate that distal lung epithelial are intimately associated with the inflammatory cells and endure within the escalating inflammatory and hypoxic environment of the advancing TB lesions. Moreover, we found that a substantial proportion of epithelial cells clustered within advanced granulomatous lesions were phenotypically similar to Nkx2.1+ dysplastic epithelial cells found in pathological conditions caused by epithelial damage and aberrant repair characterized by the bronchiolization of the alveolar epithelium and fibrosis [58–60]. For example, lung epithelial damage and repair in non-infectious models was shown to induce the accumulation of alternatively activated macrophages that supported epithelial regeneration [61,62]. We propose that the aberrant epithelial repair within TB lesions, also turns on a maladaptive lung tissue-specific repair program, including the induction and recruitment of the Arg1+ reparative macrophages. Our data indicate that Arg1 expression is initiated in Sca1+ progenitors in bone marrow and Mtb-infection induces the expansion of Arg1+CD11b+Ly6C+ monocytes committed to M2-like differentiation prior to their recruitment to PTB lesions. These findings resemble the induction of myeloid progenitors committed to M2 polarization during lung tumorigenesis [63] and explain simultaneous appearance of the distinct iNOS+ and Arg1+ macrophage populations within lesions dominated by the M1-polarizing cytokines, including IFN-I [64].

Remarkably, the alternatively activated macrophages colocalized with iNOS+ macrophages and often cloak them, similar to previously described cloaking of tissue microlesions by resident macrophages to prevent neutrophil-driven inflammatory damage [65]. Perhaps, these intimate interactions of the M1- and M2-like macrophages within PTB lesions compromise local host defenses and permit Mtb replication. Then, the lesion progression is driven by Mtb replication and takes a precipitous course towards the advanced non-controlling lesions. Indeed, the induction of Arg1+ macrophages in IL-13 transgenic mice led to the formation of necrotic PTB lesions [66].

The subcutaneous lung implant model allowed us to compare TB progression in the lung and spleen fragments implanted side-by-side under the skin. We clearly demonstrated the formation of multibacillary TB lesions in the lung, but not in spleen implants, albeit the bacteria were found in both implants. Lung implants that contained necrotic lesions with pronounced bacterial replication, also contained Arg1+ macrophages that were absent from adjacent spleen implants. Perhaps, the selective recruitment of the Arg1+ "repair" macrophages from circulation to lung tissue may account for the observed synchronization of TB lesions in native lungs and lung implants. Taken together, these findings point to a critical role of lung parenchymal cells in creating the Mtb permissive immune environment [67], and argue against high oxygenation of lung tissue as a driver of pulmonary TB progression.

Recent studies of primary TB using aerosol infection demonstrated the role of neutrophils and dendritic cells in lung epithelial damage and IFN-I hyperactivity in the *sst1*-susceptible and Sp140 knockout mice, respectively [68,69]. Lung resident macrophages were found to harbor Mtb in the resistant B6 mice [70]. In contrast, our model of slow PTB progression after hematogenous spread revealed that the majority of myeloid cells within chronic PTB lesions were represented by two populations of the recruited macrophages that displayed either the classical or the alternative activation phenotypes.

Lung epithelial cells survived within these lesions and in response to inflammatory injury and hypoxia adopted phenotypes of transitional alveolar (Krt8+) and dysplastic airway (p63+Krt17+) progenitors. They also expressed IFNβ and possibly other inflammatory mediators fueling the maladaptive macrophage polarization and thus creating a TB-promoting microenvironment (TbME). Based on these studies, we postulate a "two-hit" hypothesis of PTB susceptibility in which (1) the aberrantly activated pro-inflammatory macrophages and (2) the dysregulated lung repair cooperates to gradually carve a niche for Mtb replication. This mechanism of pulmonary TB progression may explain the stealthy transformation of the initially resistant immunocompetent individuals into susceptible Mtb spreaders.

## Limitations and future directions

Having started to provide novel insights into fundamental questions of PTB pathogenesis, our studies also pose key mechanistic questions. In particular, specific mechanisms by which epithelial cells embedded within PTB lesions modify local immunity remain largely hypothetical. Are they mediated purely by secreted cytokines and growth factors, such as CSF1, or also employ developmental regulators and require direct cell contact? What other resident lung cell populations are critically involved in generating the permissive niche? Likely, fibrotic mesenchymal cells induced by the aberrant epithelial repair pathways [59,71] could also be a source of growth factors or potent immunosuppressive cytokines, such as IFN-I, IL-10, TGFβ and others. The roles of lung resident macrophage populations, alveolar and interstitial, various lymphocyte populations, in this crosstalk also remain to be established. Of special interest are the mechanisms of the formation of myeloid progenitors committed to M2-like polarization and their affinity for the lung tissue [63].

Although no single animal model is sufficient to reproduce the complete TB spectrum, the *sst1* susceptible mice have been instrumental in dissecting mechanisms of both primary and post-primary PTB progression. This genetically defined mouse may serve for further exploration of lung-specific mechanisms of host susceptibility to virulent Mtb and for testing therapeutic interventions specifically targeting these mechanisms.

## Methods

Ethics statement Approval for human immune system mouse animal experiments was obtained from the Institutional Animal Care and Use Committee, University of Texas Medical Branch (UTMB) (protocol number 0905041). Discarded tissue specimens were obtained via a nonprofit source as approved under exemption 4 of HHS regulations (45 CFR part 46) and as previously described [72,73].

## Reagents

Fetal bovine serum (FBS) (Cat# SH30396) for cell culture medium obtained from HyClone. Middlebrook 7H9 and 7H10 mycobacterial growth media were purchased from BD Biosciences and prepared according to manufacturer's instructions. The 50 μg/mL hygromycin was used for the reporter strains (*M. tuberculosis* Erdman, SSB-GFP, smyc′::mCherry).

Primary antibodies included Anti-iNOS (Abcam, Cat# ab15323, RRID:AB_301857), Arginase-1 (Cell Signaling Technology, Cat# 93668, RRID:AB_2800207), Iba1/AIF-1 (Cell Signaling Technology, Cat# 17198, RRID:AB_2820254), CD19 (Cell Signaling Technology, Cat# 90176, RRID:AB_2800152), CD3e (DAKO, Cat# A045201-2), Recombinant Anti-CD11b [EPR1344] (Abcam, Cat# ab133357, RRID:AB_2650514), HIF-1α (Cell Signaling Technology, Cat# 48085), Caveolin-1 (Cell Signaling Technology, Cat# 3267, RRID:AB_2275453), TTF1 (Abcam, Cat# ab76013, RRID:AB_1310784), N-Terminal Pro-Surfactant Protein-C (Seven Hills Bioreagents, Cat# WRAB-9337, RRID:AB_2335890), GFP (Invitrogen, Cat# A11122, RRID:AB_221569), β-Tubulin (Cell Signaling Technology, Cat# 2128S, RRID:AB_823664), and Uteroglobin (Abcam, Cat# ab213203, RRID:AB_2650558).

Secondary and detection reagents included ImmPRESS HRP Anti-Rabbit IgG (Vector Laboratories, Cat# MP-7451, RRID:AB_2631198), ImmPRESS HRP Anti-Mouse IgG (Vector Laboratories, Cat# MP-7452, RRID:AB_2744550), Recombinant Anti-IgG1+IgG2a+IgG3 (Abcam, Cat# ab133469, RRID:AB_2910607), Goat anti-Rabbit IgG Alexa Fluor 546

(Invitrogen, Cat# A-11010), Goat anti-Rabbit IgG Alexa Fluor Plus 647 (Invitrogen, Cat# A-32733), and ChromoMap DAB Kit (Roche, Cat# 760–159).

For flow cytometry, antibodies included Anti-CD11b BUV395 (BD Biosciences, Cat# 565976, RRID:AB_2721166), Anti-Ly6C eFluor 450 (Thermo Scientific, Cat# 50-246-021), Anti-Sca-1 RB780 (BD Biosciences, Cat# 569230), Anti-Ly6G Alexa Fluor 700 (BioLegend, Cat# 127621, RRID:AB_10640452), Anti-CD19 BV605 (BioLegend, Cat# 115539, RRID:AB_11203538), Anti-CD45 Spark Blue 574 (BioLegend, Cat# 103183, RRID:AB_2904274), and Anti-Arg1 APC (Thermo Scientific, Cat# 17-369-780).

Oligonucleotides: All primers and probes were obtained from Integrated DNA Technologies. The primers used for qRT-PCR included *Arg1* Forward (FP: AAGAAAAGGCCGATTCACCT) and Reverse (RP: CATGATATCTAGTCCTGAAAGG), *Chil3* FP (TGCCTTTGCTGGAATGCAGA) and RP (AGAGACCATGGCACTGAACGG), *Il6* FP (CCAGAGTCCTTCAGA-GAGATACA) and RP (AATTGGATGGTCTTGGTCCTTAG). Primers for *Ifnb1, Rsad2, Trib3, Chac1, β-actin*, and *18S* were used as described in Bhattacharya et al., 2021 [41].

## Animals

C57BL/6J mice were obtained from the Jackson Laboratory (Bar Harbor, Maine, USA). The B6J.C3-*Sst1*$^{C3HeB/Fej}$ Krmn congenic mice (B6.Sst1S) were created by transferring the sst1 susceptible allele from the C3HeB/FeJ mouse strain onto the B6 (C57BL/6J) genetic background via twelve backcrosses. These mice are referred to as B6.Sst1S in the text.

The B6.Sst1S,*Ifnb1*-YFP mice were produced by breeding B6.Sst1S mice with a reporter mouse containing a Yellow Fluorescent Protein (YFP) reporter gene inserted after the *Ifnb1* gene promoter [74]. The YFP serves as a substitute for Interferon beta (*Ifnb1*) gene expression. All animal experiments were approved by the Institutional Animal Care and Use Committee (IACUC) at Boston University, which is an Association for Assessment and Accreditation of Laboratory Animal Care International (AAALAC)-accredited.

## Generation of mice with lung and spleen implant

The detailed protocol is described in [46]. Briefly, Lung and spleen tissues were harvested from young syngeneic donor mice and cut into 5–10 mm grafts, which were then placed in sterile culture medium. Mice were anesthetized, the surgical site was prepared, and subcutaneous pockets were created for graft implantation. Matrigel-coated lung grafts were inserted into one pocket, while spleen grafts were placed in the contralateral side. Post-surgery, mice received analgesics and were monitored for recovery.

## Infection of mice and collection of organs

The mice were anesthetized using ketamine-xylazine solution administered by intraperitoneal route. Each mouse was injected subcutaneously in the hock, the lateral tarsal region (by placing animal in the restrainer) just above the ankle with 50 µl of the 1X PBS containing $10^6$ CFU of Mtb H37Rv or Mtb Erdman(SSB-GFP, smyc′::mCherry [48]). At desired time points the mice were anesthetized, lung perfusion performed and organs collected.

## Tissue inactivation, processing, and histopathologic interpretation

Tissue samples were submersion fixed for 48 h in 10% neutral buffered formalin, processed in a Tissue-Tek VIP-5 automated vacuum infiltration processor (Sakura Finetek, Torrance, CA, USA), followed by paraffin embedding with a HistoCore Arcadia paraffin embedding machine (Leica, Wetzlar, Germany) to generate formalin-fixed, paraffin-embedded (FFPE) blocks, which were sectioned to 5 µm, transferred to positively charged slides, deparaffinized in xylene, and dehydrated in graded ethanol. A subset of slides from each sample were stained with hematoxylin and eosin (H&E) and consensus qualitative histopathology analysis was conducted by two board-certified veterinary pathologists (N.A.C. & M.L.) to characterize the overall heterogeneity and severity of lesions.

## Quantitative image analysis (pneumonia classifier and multiplex fluorescent and chromogenic immunohistochemistry)

Digitalized whole slide images (WSI) were generated for affiliated brightfield and multiplex fluorescent immunohistochemistry (mfIHC) panels were generated using a multispectral Vectra Polaris Automated Quantitative Pathology Image System (Akoya Biosciences, Marlborough, Massachusetts, USA). Quantification and classification of the pulmonary inflammatory lesions were achieved by using the HALO image analysis software v3.5 (Indica labs, Albuquerque, New Mexico, USA) random forest V2 tissue classifier. Each test image was first annotated by pathologists to define the regions of interest for subsequent image analysis. Total examined pulmonary parenchyma with associated airways were included in the regions of interest. All non-pulmonary tissues such as heart, esophagus, tracheobronchial lymph nodes, mediastinal adipose tissues, peripheral nerves, major vessels, and extra pulmonary bronchi with cartilage were excluded from the regions of interest. All cells and tissues outside the visceral pleura of the lung lobes and processing artifacts (I.e., atelectic pulmonary parenchyma resulting from insufficient postmortem lung inflation, tissue folds and dust) were also manually removed via exclusion annotations. Two classes were defined: normal and pulmonary inflammation (consolidation). The algorithm was trained on five representative cases of B6.Sst1S infected mice. The full diversity of pulmonary inflammatory spectrum was included in training, which included granulomatous pneumonia, necrosuppurative pneumonia, perivascular and peribronchiolar mononuclear infiltrates. Representative classification images are displayed in S1A Fig. The exact same classifier algorithm was applied to all cases in this study to yield the pulmonary consolidation quantification results.

For quantification of both chromogenic and fluorescent IHC assays using positive pixel analysis, representative areas of granulomatous pneumonia and normal parenchyma of each animal analyzed were annotated to define the regions of interest for analysis. This approach was utilized to quantify the positive immunoreactivity area for myeloid markers (CD11b, iNOS, Arg1, and Iba1) and HIF-1α in different stages of granulomatous pneumonia across timepoints and genotypes.

For immunophenotyping and quantification of cell density the Halo HighPlex FL v4.2.3 algorithm was utilized. mfIHC slides were scanned and subsequently multispectrally unmixed using Inform software (Akoya Biosciences) using multispectral libraries. Representative areas of granulomatous pneumonia and normal parenchyma of each animal were annotated to define the regions of interest. Animals with necrosuppurative pneumonia were excluded in this analysis due to extensive necrosis in the lesions. This approach was utilized to determine the density of Nkx2.1(Ttf-1) and SPC single and double positive cells in the areas of interest, as well as T and B cell densities.

## Mycobacterial staining of lung sections

To visualize mycobacteria in Mtb infected lung specimens, the brightfield acid-fast staining or auramine O-rhodamine B methods were utilized. For the brightfield acid-fast staining of Mtb infected lung sections, the new Fuchsin method (Poly Scientific R and D Corp., cat no. K093, Bay Shore, NY, USA) was used according to the manufacturer's instructions. The auramine O-rhodamine B method was performed with slight modifications to a previously described protocol [23]. The sections were deparaffinized and rehydrated through graded ethanol washes (100%, 95%, 70%, 50%, 25% and distilled water), after which they were stained with auramine O-rhodamine B at 37°C for 15 minutes. Excess stain was removed by washing the sections in 70% ethanol three times for 1 min each. The sections were counterstained with Mayer's hematoxylin for 5 min and washed with water. Finally, the sections were dehydrated and mounted with a permount mounting medium.

To determine whether lesion formation was associated with non-acid-fast forms of Mtb, a dual-staining approach combining immunohistochemistry (IHC) and acid-fast staining was employed. Lung lesion sections from Mtb-infected mice were initially subjected to IHC using a polyclonal anti-mycobacterial antibody (Biocare Medical, Cat# CP140A) to detect mycobacterial antigens. Subsequently, the same sections were stained using the New Fuchsin method to identify acid-fast bacilli (details provided above). This approach allowed the differentiation between acid-fast bacilli and non-acid-fast mycobacterial forms within the same lesion.

## Semi-quantification of acid-fast bacilli (AFB) load

Mtb load in the lung and other tissues (I.e., spleen, popliteal lymph node, liver, and large intestines) was first qualitatively examined and subsequently an ordinal scoring system was developed to capture the overall variability in bacterial load. The ordinal criteria for the semi quantification was defined as the following: + indicates rare, individualized acid-fast bacilli (AFB) (either less than or equal to 1 per 200X field in the most severely impacted region; or there are only one to two 200X fields containing two to three individualized AFB throughout the entire section); ++ indicates sporadic single AFB with occasional chains to small clusters (more than two 200X fields contains more than 3 AFB, but AFB are not easily visible in 200X field); +++ indicates frequent clusters AFB (easily visible at 200x and even visible at 100x); ++++ indicates innumerable AFB with large clusters easily visible at 40x. S1C Fig showed representative images of Mtb load + to ++++.

## Confocal immunofluorescence microscopy of cleared lung tissue sections

The detailed protocol is described in [53]. Briefly, fixed lung lobes were embedded in a 4% agarose solution in 1X phosphate-buffered saline (PBS), and 50 µm thick sections were obtained using a Leica VT1200S vibratome. The lung sections were permeabilized by treatment with 2% Triton X-100 for 24 hours at room temperature. After permeabilization, the lung sections were washed three times for 5 minutes each with 1X PBS and then blocked in a solution of 1X PBS containing 3% bovine serum albumin (BSA) and 0.1% Triton X-100 for 1 hour at room temperature. Subsequently, the sections were incubated with primary antibodies overnight at room temperature. After primary antibody incubation, the samples were washed three times for 5 minutes each with the blocking buffer and then incubated with secondary antibodies at room temperature for 2 hours, with all steps performed on a nutator. Following the secondary antibody incubation, the samples were washed three times for 5 minutes each with the blocking buffer, stained with Hoechst 33342 solution (Fisher Scientific) for nuclei detection, washed three times for 5 minutes each with the blocking buffer, and finally transferred into RapiClear 1.47 solution (Sunjin Lab, Taiwan, catalog #NC1660944) for tissue clearing at room temperature on a nutator for 24 hours. The primary antibodies anti-Nkx2.1, anti-iNOS, anti-αSMA and anti-Col1a1 were used at a dilution of 1:100, while Alexa Fluor 647 goat anti-rabbit antibody (from Invitrogen) was used at a dilution of 1:200 for detection.

All images were acquired using a Leica SP5 confocal microscope, and image processing was performed using Imaris Viewer (from Oxford Instruments) and ImageJ software (from NIH).

## RNA Isolation from Mtb infected mice blood and quantitative PCR

To isolate RNA from Mtb-infected mice, 100 µL of blood was collected in a microcentrifuge tube containing 200 µL of RNAlater Stabilization Solution (Invitrogen, Cat# AM7020). The samples were centrifuged, and the supernatant was removed. The pellet was lysed in 1 mL of TRIzol reagent (Invitrogen, Cat# 15596018). Total RNA extraction was performed using the RNeasy Plus Mini Kit (Qiagen). cDNA synthesis was then carried out using the PrimeScript RT Master Mix (Takara, Cat# RR036A). Real-time PCR was conducted with GoTaq qPCR Mastermix (Promega) using the CFX-96 system (Bio-Rad). Primers were designed using Integrated DNA Technologies, with 18S as internal controls, and fold induction was calculated using the ddCt method.

## Spatial transcriptomic analysis of TB lesions

To perform spatial transcriptomics analysis, we used the Nanostring GeoMX Digital Spatial Profiler (DSP) system (Nanostring, Seattle, WA) [75,76]. To identify pathways dominating early vs. late (advanced) lesions, we selected lungs from 2 mice with paucibacilarly early lesions without necrosis and 2 mice with advanced multibacilarly lesions with necrotic areas. Slides were stained with fluorescent CD45-, and pan-cytokeratin-specific antibodies, and DAPI. Diseased regions of interest (ROI) were selected to focus on myeloid-rich areas avoiding areas of micronecrosis and tertiary lymphoid tissue. Eight ROI each of normal lung, early and late lesions (respectively) were studied. The profiling used the Mouse Whole

Transcriptome Atlas (WTA) panel which targets ~21,000+transcripts for mouse protein coding genes plus ERCC negative controls to profile the whole transcriptome, excluding uninformative high expressing targets such as ribosomal subunits. Samples from each ROI were packaged into library for sequencing (NextSeq550, Illumina) following the procedure recommended by Nanostring. After sequencing, the data analysis workflow began with QC evaluation of each ROI based on thresholds for number of raw and aligned reads, sequencing saturation, negative control probe means, and number of nuclei and surface area. Background correction is performed using subtraction of the mean of negative probe counts. Q3 normalization (recommended by Nanostring) results in scaling of all ROIs to the same value for their 3rd quartile value. The count matrix was imported into the Qlucore Genomics Explorer software package (Qlucore, Stockholm, Sweden) and log2 transformed for further analysis. Statistical analysis was then performed to identify differentially expressed genes (typically unpaired t-test with Benjamini-Hochberg control for FDR rate at q < .05). Lists of differentially expressed genes (DEGs) from comparisons of distinct stages were further analyzed for enrichment of pathways or predicted transcription factors using the Enrichr online tool [77–79]. The GSEA analysis tool of the Qlucore software was used to evaluate enrichment of the MSigDB Hallmark collection of pathways [79]. Spatial transcriptomics data was deposited to GEO dataset GSE292392 (https://www.ncbi.nlm.nih.gov/geo/query/acc.cgi?acc=GSE292392).

### Processing samples for blood transcriptomics and data analysis

Transcriptome profiling of RNA samples isolated as above was performed using the mouse DriverMap assay (Cellecta, Inc, Mountain View, CA), a single-tube RT-PCR/ Next-Generation sequencing assay that robustly measures expression levels of ~4800 mouse protein-coding genes from ng levels of total RNA, as detailed in https://www.manula.com/manuals/cellecta/drivermap-exp-hgw19k-v3-kit/v1/en/topic/drivermap-air-rna-profiling-assay. After sequencing, gene expression differences were evaluated using the iDEP web platform (https://bioinformatics.sdstate.edu/idep/)and EdgeR normalization [80]. An adjusted p-value of <.05 was used to define significant DEGs. Pathway analysis was peformed using the Enrichr tool [78] along with entry of the list of DriverMap genes evaluated as the background.

### Flow cytometry

Bone marrow cells were isolated from uninfected and infected mice. RBC lysis was performed using RBC lysis buffer (eBioscience Cat. No. 00-4300-54) following the instructions provided. The cells were stained for dead cells using Zombie NIR Fixable Viability Kit (Biolegend Cat No. 423105). The blocking was performed using TruStain FcX PLUS (anti-mouse CD16/32 antibody) (Biolegend Cat. No. 156603) and monocyte blocker (Biolegend Cat No. 426101) followed by surface staining for 30 min. The antibodies used for surface staining were listed in S6 Table. Further, cells were fixed and permeabilized using Intracellular Fixation & Permeabilization Buffer Set (eBioscience Cat# 88-8824-00). Intracellular staining was performed for Arg1, and cells were analyzed by Cytek Aurora full spectrum flow cytometer (UV/V/B/ YG/R). For single stain fluorescence reference controls, bone marrow cells and Ultracomp eBeads Plus (Thermo Fisher Cat. No. 01-3333-42) were stained with each antibody and viability dye. Spectral unmixing and autofluorescence extraction was carried out using Cytek SpectroFlo software v3.2.1 unmixing algorithm. The unmixed data was analyzed using FlowJo software v10.10.0. The population frequency data was plotted and analyzed using Prism 8 software (GraphPad).

### Generation of human lung xenograft mouse model

Six NOD-SCID/γc(null) (NSG) mice (Jackson Laboratories), approximately 3–5 weeks of age, were engrafted with human lung specimen fragments as previously described [73]. In brief, lung fragments were sutured to muscle fascia in the dorsal subcutaneous space in each mouse (~0.5 cm from the spine). Simultaneously, the bone marrow, liver, thymus (or BLT) engraftment of a human immune system was performed using tissue specimens and CD34+human cells as described [72,73]. Animals received appropriate post-surgery treatment including antibiotics and analgesics.

## Infection process and sample preparation

Following 12 wk of engraftment, animals were transferred to a biosafety level 3 (BSL3) housing for experiments with Mtb. Animals were infected with 200 CFU of Mtb H37Rv in 40 µl of PBS (20 µl/nare) through an intranasal route as described [81]. Mouse lung and human lung explants were harvested during necropsy and specimens were fixed by using 10% neutral buffered formalin for 7 days under BSL3 conditions including replacement of fresh 10% formalin following 2 days.

## Statistical analysis

To compare multiple groups with two or more variables, we employed a two-way analysis of variance (ANOVA) with adjustments made for multiple post hoc comparisons. Various comparisons were conducted in our study, including comparisons among all groups, between the control group and all other groups, and between selected groups. When comparing multiple groups with only one variable, we utilized a one-way ANOVA and corrected for multiple post hoc comparisons. In cases where only two groups were being compared, two-tailed paired or unpaired t-tests were employed. The Log-rank (Mantel-Cox) test was used to determine the statistical significance in mouse survival data. All statistical analyses were performed using GraphPad Prism 9 software. We considered a p-value < 0.05 as statistically significant. Statistical significance is denoted using asterisks (*, $P < 0.05$; **, $P < 0.01$; ***, $P < 0.001$; ****, $P < 0.0001$).

## Supporting information

**S1 Fig. Quantification of areas of inflammation and Mtb bacterial loads. A**. Representation for B6.Sst1S lesions quantification by digital classifier. Representative whole slide images of lungs of individual animals with severe pneumonia, moderate pneumonia, and lung that is within normal limits (Top panel, left to right, respectively). The bottom panel shows corresponding toggled images with areas classified as consolidation (pneumonia) by HALO image analysis software labeled in red. Note the absence of red labeling in the lung that is within normal limits (the right panel images), indicating the specificity of digital classification. **B**. Representative histopathology and AFB load of infected C57BL/6J mice at 11 and 20 weeks post infection. Left two columns: H&E stain, 100X. Right two columns: Ziehl- Neelsen acid fast stain, 400X. **C**. Representation of criteria of semi quantification of Mtb load in the lungs. Left upper, right upper, left lower, and right lower image represent Mtb loads of +, ++, +++, and ++++, respectively. Please refer to the material and methods for the detailed description of the criteria. **D** - **E**. Areas of complete tissue necrosis and the basophilic amorphous material adjacent to the thrombosed medium-caliber vessel (dystrophic mineralization), H&E. **F**. The bronchioles are often occluded with large numbers of cell debris containing neutrophils and macrophages overflowed from the consolidated alveolar spaces. Occasional thrombi are observed in the lesions as shown in the center and 7 o'clock of the image, H&E. **G**. Fibrinoid necrosis of a vessel in the area of necrosis. H&E.
(TIF)

**S2 Fig. Systemic dissemination of Mtb.** A. Representative images of histopathology (left) and acid-fast bacilli (right) staining in the lungs of *M. tuberculosis* infected B6.Sst1S mice at 6 weeks post hock infection, 200X original magnification.**B**. Representative histopathology and AFB load in popliteal lymph node of B6.Sst1S mouse at 6 weeks post hock infection with $10^6$ CFU of Mtb Erdman. Left panel - clusters of macrophages (microgranulomas) scattered in the peri-trabecular and medullary sinuses of popliteal lymph node (H&E stain, 100X). Middle – microgranuloma (H&E stain, 400X). Right panel - Ziehl-Neelsen acid fast stain, 400X. A few acid-fast bacilli (arrowhead) are scattered in the clusters of macrophages. **C**. Representative images of histopathology and acid-fast bacilli load in the spleen of *M. tuberculosis* infected B6.Sst1S mice at 20 weeks post hock infection. Left image: clusters of macrophages (microgranulomas) scattered in the white pulp. H&E stain, 100X. Middle image: higher magnification of microgranulomas in the white pulp (H&E stain, 200X). Right image: A few single acid-fast bacilli (arrow) are scattered in the clusters of macrophages in the white pulp. Ziehl-Neelsen acid fast stain, 400X. **D**. Representative images of histopathology and acid-fast bacilli load in the liver

of *M. tuberculosis* infected B6.Sst1S mice at 20 weeks post hock infection. Left image: a rare cluster of macrophages in the sinusoids. H&E stain, 200X. Right image: A few single acid-fast bacilli (arrow) are scattered in the clusters of macrophages in the sinusoids. **E.** Representative images of histopathology and acid-fast bacilli load in the cecum of *M. tuberculosis* infected B6.Sst1S mice at 20 weeks post hock infection. Left image: multifocal small clusters of macrophages (microgranulomas) scattered in the GALT (gut- associated lymphoid tissues). H&E stain, 100X. Middle image: higher magnification of microgranulomas in the GALT (H&E stain, 200X). Right image: A few single acid-fast bacilli (arrow) are scattered in the clusters of macrophages in the GALT. Ziehl-Neelsen acid fast stain, 400X. **F**. Histopathological scores indicating Mtb loads across different organs in B6 (n = 9) and B6.Sst1S (n = 32) mice. Each mark on the X-axis represents an individual mouse.
(TIF)

**S3 Fig. Comparative Analysis of Paucibacillary and Multibacillary TB Lesions in B6 and B6.Sst1S Mice. A.** Duplex immunohistochemistry (Mtb polyclonal) combined with Acid Fast Bacilli (AFB) histochemical stain. Paucibacillary (P) and multi-bacillary (M) lesions. Mtb antigen and AFB colocalized within identical anatomical compartments (i.e., cytoplasm of macrophages). Images clearly show that Mtb bacilli and antigen loads were low in C lesions, significantly increasing in NC lesions. All images taken at 400X magnification. **B.** 3D confocal images of 50 μm thick cleared lung sections of paucibacillary and multi- bacillary intermediate and terminal TB lesions of mice infected with Mtb reporter strain expressing replication marker (SSB-GFP, *smyc'*::mCherry). Replication reporter (SSB-GFP) is shown as green dots; Mtb (smyc':: mcherry) is red and nuclei in grayscale (DAPI). The fraction of the replication reporter-positive Mtb in intermediate M lesions increased to 30–50% (left and middle panels). Scale bar- 50μm. **C**. fmIHC multiple images for wild type B6, B6.Sst1S- paucibacillary (P), and B6.Sst1S- multi-bacillary (M) TB lesions at 40X magnification showing accumulation of activated macrophages (iNOS) in B6.Sst1S- (M), with dissolution of discrete lymphoid follicles. DAPI- Gray; CD19 - green; CD3 epsilon (CD3ε) - red; inducible nitric oxide (iNOS) -teal; ionized calcium-binding adaptor molecule 1 (Iba1) - blue. All images taken at 10X magnification.
(TIF)

**S4 Fig. Blood transcriptome analysis of circulating myeloid cells. A - B.** Volcano plots showing differentially expressed genes in blood samples from M. tuberculosis-infected mice compared to non-infected controls (A) and blood samples from M. tuberculosis-infected mice with paucibacillary lung lesions compared to non-infected controls (B). Each point represents a single gene, with the x-axis indicating log2 fold change and the y-axis showing -log10(p-value). Significantly downregulated and upregulated genes are highlighted in red (left and right, respectively), while non-significant genes are shown in gray. Significance thresholds are set at p < 0.05 and fold change > ±1.5. **C.** Expression of Trib3 mRNA in blood of noninfected control and Mtb infected mice with multi-bacillary stage lung lesions. **D - E.** Quantification of M1 and M2 macrophage-associated genes, as determined by blood transcriptomics, in mice with paucibacillary and multibacillary TB lung lesions compared to non-infected controls, showing overlap and distinctions in gene expression associated with each macrophage polarization state. The statistical significance was performed by two-tailed unpaired t test (Panel C). Significant differences are indicated with asterisks (*, P < 0.05; **, P < 0.01; ***, P < 0.001; ****, P < 0.0001).
(TIF)

**S5 Fig. Mtb infection leads to increase in Arg1 expressing cells in bone marrow. A.** Gating strategy for flow cytometry data analysis. **B.** The CD45 vs Arg1 plot showing the increase in Arg1 expressing cells in bone marrow upon Mtb infection, five mice of each group.
(TIF)

**S6 Fig. Dysplastic lung epithelial cells in PTB lesions of B6.Sst1S mice. A.** Comparison of representative images of chromogenic immunohistochemistry staining of Nkx2.1 (magenta), Surfactant Protein C (SPC, brown) and Caveolin-1

(green) in unaffected pulmonary parenchyma in a B6 mouse versus granulomatous pneumonia lesions in a B6.Sst1S, *Ifnb1*-YFP mouse. **B.** Representative images of chromogenic immunohistochemistry staining of Nkx2.1 (magenta), C (SPC, brown) and Caveolin-1 (green) of necrosuppurative pneumonia of Mtb infection in a B6.Sst1S,*Ifnb1*-YFP mouse (top panel) and granulomatous pneumonia in a B6.Sst1S mouse (bottom panel). Left panel-top: 100X; bottom: 200X. Right panel- Corresponding toggled digital classification of the images in the left panel showing indicative IHC signals recognized by the trained algorithm (Nkx2.1 recognized in red, SPC recognized in green and Caveolin-1 recognized in blue). (TIF)

**S7 Fig. The expression of the IFNβ reporter YFP in TB lesions. Majority of YFP/IFNβ+ cells are iNOS+ activated macrophages. A.** Additional cell populations illustrated to express GFP/IFN reporter (iNOS+ macrophages- left; perivascular Iba- mononuclear cells –middle; airway epithelium-right). **B.** 3D confocal images of cleared 50 μm thick lung sections of B6.Sst1S,*Ifnb1*-YFP and (C3H x B6.Sst1) F1 mice. Endogenous YFP (seen only in B6.Sst1.S,*Ifnb1*-YFP) is shown in green, iNOS staining in teal blue, reporter bacteria in red (smyc':: mcherry) and nuclei in grayscale (DAPI). Scale bar- 50μm. **C.** 3D confocal image of cleared 50 μm thick lung sections of B6.Sst1S,*Ifnb1*-YFP mice stained with Col1a1 antibody; on the right – individual images of Col1a1+cells adjacent to YFP (denoted by white boxes on the left panel). Scale bar- 50μm. **D.** 3D confocal images of cleared 50 μm thick lung sections of B6.Sst1S,*Ifnb1*-YFP mice stained with myofibroblast marker αSMA antibody. Endogenous YFP is shown in green, αSMA staining in magenta, and nuclei in grayscale (DAPI); on the right – individual images show that αSMA+ cells are adjacent to YFP (denoted by white boxes on the left panel). Scale bar- 50μm. (TIF)

**S8 Fig. Characterization of TB lesions in lung and spleen implants. A.** Correlation of acid-fast bacilli (AFB) score in lungs and respective lung implants of mtb- infected mice. AFB score were annotated via Ziehl-Neelsen staining and microscopic observations in both primary lung tissues and implants. **B.** Representative hematoxylin and eosin (H&E) and acid-fast bacilli (AFB) staining of spleen implant. 200x total magnification. **C.** fmIHC images of Arg1+ and iNOS+ cells in lung and spleen implants of Mtb infected mice. Arg1+cells - orange color and iNOS+- teal color. **D.** fmIHC images of Caveolin-1+cells (left) at low magnification and Nkx2.1+ and Caveolin-1+cells (right) at high magnification in lung and spleen implants of Mtb infected mice. **E.** Corresponding TB lesions in the native lung of B6.Sst1S recipient mice (upper panel) and in implanted lung tissue isolated from B6 donors (lower panel). Implanted B6 lung tissue developed the necrotic lesions containing numerous Mtb. H&E and AFB staining. (TIF)

**S1 Table. Histopathology and acid-fast bacilli load in lung of all animals.**
(DOCX)

**S2 Table. Histopathology and acid-fast bacilli load in spleen of examined animals.**
(DOCX)

**S3 Table. Histopathology and acid-fast bacilli load in popliteal lymph nodes of examined animals.**
(DOCX)

**S4 Table. Differentially expressed genes (DEGs) in total ROIs in paucibacillary vs multibacillary TB lesions.**
(XLSX)

**S5 Table. Pathways upregulated in multibacillary vs paucibacillary TB lesions (Total ROIs).**
(DOCX)

**S6 Table. The list of antibodies used for staining cells for flow cytometry.**
(DOCX)

**S7 Table. Cell counts of Arg1 expressing cells in bone marrow.**
(DOCX)

## Acknowledgments

We are grateful to Drs. Roderick Bronson and Anatoly S. Gleiberman for helpful discussions of lung regeneration and carcinogenesis. The authors are grateful for the expert support of the Boston University Flow Cytometry Core Facility and the NEIDL Comparative Pathology Laboratory (NCPL).

## Author contributions

**Conceptualization:** Shivraj M. Yabaji, Darrell Kotton, Janice J. Endsley, William R Bishai, Lester Kobzik, Igor Kramnik.

**Data curation:** Shivraj M. Yabaji, Lester Kobzik.

**Formal analysis:** Shivraj M. Yabaji, Suruchi Lata, Anna E. Tseng, Prasanna Babu Araveti, Ming Lo, Anna C. Belkina, Hirofumi Kiyokawa, Nicholas Crossland, Lester Kobzik.

**Funding acquisition:** Igor Kramnik.

**Investigation:** Shivraj M. Yabaji, Suruchi Lata, Prasanna Babu Araveti, Ming Lo, Igor Gavrish, Nicholas Crossland, Lester Kobzik, Igor Kramnik.

**Methodology:** Shivraj M. Yabaji, Suruchi Lata, Anna E. Tseng, Prasanna Babu Araveti, Ming Lo, Igor Gavrish, Aoife K O'Connell, Hans P Gertje, Anna C. Belkina, Colleen E Thurman, Shumin Tan, Janice J. Endsley, Nicholas Crossland.

**Project administration:** Igor Kramnik.

**Resources:** Shumin Tan, Nicholas Crossland, Lester Kobzik, Igor Kramnik.

**Software:** Igor Kramnik.

**Validation:** Shivraj M. Yabaji, Suruchi Lata, Anna E. Tseng, Prasanna Babu Araveti, Ming Lo, Anna C. Belkina, Hirofumi Kiyokawa, Nicholas Crossland, Lester Kobzik.

**Visualization:** Suruchi Lata, Janice J. Endsley, Nicholas Crossland, Igor Kramnik.

**Writing – original draft:** Shivraj M. Yabaji, Nicholas Crossland, Lester Kobzik, Igor Kramnik.

**Writing – review & editing:** Hirofumi Kiyokawa, Darrell Kotton, Shumin Tan, Janice J. Endsley, William R Bishai, Nicholas Crossland, Lester Kobzik, Igor Kramnik.

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
