## [Decision Letter · Decision Letter 0]

11 Aug 2025

PPATHOGENS-D-25-01461

Dysplastic lung repair fosters a tuberculosis-promoting microenvironment through maladaptive macrophage polarization

PLOS Pathogens

Dear Dr. Kramnik,

Thank you for submitting your manuscript to PLOS Pathogens. After careful consideration, we feel that it has merit but does not fully meet PLOS Pathogens's publication criteria as it currently stands.

Specifically, both reviewers noted the importance and novelty of the work.  Please address the major concerns of reviewer #1.

We  invite you to submit a revised version of the manuscript that addresses the points raised during the review process.

Please submit your revised manuscript within 60 days Oct 10 2025 11:59PM. If you will need more time than this to complete your revisions, please reply to this message or contact the journal office at plospathogens@plos.org. Please include the following items when submitting your revised manuscript:

We look forward to receiving your revised manuscript.

Kind regards,

David M. Lewinsohn

Academic Editor

PLOS Pathogens

Anne Jamet

Section Editor

PLOS Pathogens

Sumita Bhaduri-McIntosh

Editor-in-Chief

PLOS Pathogens

orcid.org/0000-0003-2946-9497

Michael Malim

Editor-in-Chief

PLOS Pathogens

orcid.org/0000-0002-7699-2064

**Journal Requirements:**

At this stage, the following Authors/Authors require contributions: Shivraj M. Yabaji, Suruchi Lata, Anna E. Tseng, Prasanna Babu Araveti, Ming Lo, Igor Gavrish, Aoife K O’Connell, Hans P Gertje, Anna C. Belkina, Colleen E Thurman, Hirofumi Kiyokawa, Darrell Kotton, Shumin Tan, Janice J. Endsley, William R Bishai, Nicholas Crossland, Lester Kobzik, and Igor Kramnik. Please ensure that the full contributions of each author are acknowledged in the "Add/Edit/Remove Authors" section of our submission form.

- ® on page: 27

- TM on pages: 14, 22, 27, and 29.

6)  Please ensure that the funders and grant numbers match between the Financial Disclosure field and the Funding Information tab in your submission form. Note that the funders must be provided in the same order in both places as well.  

7) Thank you for stating 'Spatial transcriptomics data was deposited to GEO dataset GSE292392 (https://www.ncbi.nlm.nih.gov/geo/query/acc.cgi?acc=GSE292392).' Please note that, though access restrictions are acceptable now, your entire minimal dataset will need to be made freely accessible if your manuscript is accepted for publication. This policy applies to all data except where public deposition would breach compliance with the protocol approved by your research ethics board. If you are unable to adhere to our open data policy, please kindly revise your statement to explain your reasoning and we will seek the editor's input on an exemption.

8) Kindly revise your competing statement to align with the journal's style guidelines: 'The authors declare that there are no competing interests.'

**Reviewers' Comments:**

Reviewer's Responses to Questions

**Part I - Summary**

Reviewer #1: In this study, Yabaji et al. characterize pulmonary lesions formed after high dose intradermal Mtb infection and subsequent dissemination in B6.Sst1S mice. Their broad characterization describes key findings: 1) that these lesions exhibit heterogenous structures that span much of the spectrum that has been previously described via aerosol infection in this model; 2) Small lesions which contain fewer bacilli are typified by lymphoid aggregates and iNOS+ macrophages with an “M1” transcriptional signature, while large lesions with more bacilli are more enriched for Arg1+ macrophages with an “M2” transcriptional signature (associated with Arg1+ precursors in the marrow) and contain neutrophil-rich necrosis; 3) lesions are closely associated with epithelium, which produces inflammatory mediators including IFNb and that the lung parenchyma significantly influences the types of lesions and cellular activation in the setting of Mtb infection.

Enthusiasm for this manuscript is somewhat dampened by the need for further clarity of what this high dose intradermal inoculum reflects, largely correlative work from processes present at late timepoints, and the need for additional clarity about the robustness of the findings within animals and across experiments.

Reviewer #2: The authors of this manuscript address an important issue in Mycobacterium tuberculosis (Mtb) research – the mechanisms underlying the development of tissue-damaging pathology in host lung tissues infected with the tubercle bacillus as a result of hematogenous seeding. This is highly significant as this mode of spread of Mtb to the lungs, purportedly a route operative in post-primary TB, the major cause of severe pulmonary pathology that can promote transmission of the tubercle bacillus and compromise respiratory functions even with curative therapy. Post-primary TB is an entity that is not well studied, and as a result, the pathogenesis of this form of infection is not well understood. As a result, it is difficult to model.

The studies employ the B6.Sst1S mouse model that was originally developed by Igor Kramnik. This mouse strain develops pulmomic tuberculous lesions that are, arguably, most akin to that observed in the lungs of human infected with Mtb. In order to generate a scenario that reproduces hematogenous spread of Mtb within an infected (a route associated with post-primary TB), the authors use a variation of the prototypic B6.Sst1S model by altering the route of infection (subcutaneous hock injection as opposed to aerosolization). The authors also include a study involving lung tissue implantation in order to probe the intrinsic features of this organ in fostering susceptibility to Mtb and promoting the development severe pathology. The pathology of lung TB in these B6.Sst1S mice are evaluated with a compendium of methods including quantitative fluorescent and chromogenic immunohistochemistry analysis of tissues, in conjunction with specialized software to target specific immune cells and the lung epithelium; tissue spatial transcriptomics; blood transcriptomics, flow cytometry; and a human-like model by implanting human lung tissues into tuberculous humanized NSG mice (NOD-SCID/�c(null) engrafted with bone marrow, liver, and thymus to reconstitute a human-like immune system. Although, as pointed out by the authors, no animal models can precisely reproduce human TB, the tractable B6.Sst1S variant model described in the manuscript (and now well established as a result of the studies described in the manuscript) will prove useful for the dissection of the mechanisms by which severe pulmonary pathology is developed in TB, as in the case of post-primary infection, which involves hematogenous seeding to the lungs.

The authors are familiar with the model (Kramnik developed the B6.Sst1S strain and have been studying them to understand TB pathogenesis) and the methods used. The manuscript is well written and clearly presented, the experimental approach is rigorous and stringent, and the interpretation of the data logical and justified. Based on the results presented, the authors concluded that i) pulmonic tissues are uniquely susceptible to Mtb, independent of the relatively higher oxygen tension in the lungs, to promote bacterial replication; rather, ii) the nature of the pulmonic parenchyma is conducive to the high susceptibility of the lungs to Mtb; iii) the tuberculous lesions in the lungs of the hematogenously seeded susceptible B6.Sst1S mouse display a unique immunological environment that can support the development of severe pathology, including the co-existence of canonically and alternatively activated macrophages that are located in a specific anatomic relationship, with the Arg1+ M2 type cells cloaking the iNOS+ M1 type macrophages, perhaps masking interaction between the inflammatory macrophages with T cells; iv) the existence of dysplastic lung epithelial cells that is an integral part of the structure of the multibacillary lesions in the B6.Sst1S mouse suggests that the lung epithelial could play a role in the development of severe pulmonic pathology; v) the lesions manifesting a lymphocytic response may contribute to better control the tubercle bacillus (per bacterial load), relative to those displaying a myeloid cell reaction (particularly macrophages and neutrophils); vi) the B6.Sst1S results can be observed in implanted human lung tissues and the result of a humanized mouse model treks that observed in the mouse.

The uniqueness of this work is the modified B6.Sst1S model with a human component; the extensive, stringent, and rigorous analyses of lung tissues infected with Mtb via the hematogenous route using a compendium of experimental approaches; and the feature of tractable sequential development of increasingly severe lung pathology as the infection progresses. The studies thus provide a wealth of information that will open up multiple lines of investigation that will likely shed light on the development of TB pathology, a process that is pivotal to mitigating transmission of the tubercle bacillus and the development of pulmonary function sequalae. Particularly of interest is the role of the alternatively activated Arg1+ macrophage population and the presence of dysplastic lung epithelial within the granulomatous lesions.

**Part II – Major Issues: Key Experiments Required for Acceptance**

Reviewer #1: How much of the observed pathology is related to the dose of Mtb given, and what are the implications of this for translational relevance? In other published studies of infection with 10^4 Mtb in B6 mice (Kupz et al. 2016, Nemeth et al. 2020), there is little to no CFU present in the lung. This is in contrast to these findings showing scattered lesions with 10^6 Mtb. Additionally, infection with 10^4 Mtb in C3HeB/FeJ (a different genotype, albeit with similar pathology) mice did not show necrotic lesions (39007152), in contrast to these findings in B6.Sst1S mice. Could it be that this high dose in the dermis leads to rapid hematogenous dissemination prior to the onset of adaptive immunity (at least in some mice)? In such mice, lesion necrosis could occur prior to local adaptive immunity. This is critical to the authors’ conclusions and interpretation of what this model reflects. The authors should consider these possibilities in their model and discussion. A time course showing the kinetics of infection over time may shed more light onto this physiology.

Given the lesion heterogeneity that is well described (largely by careful analysis by many of these authors) in B6.Sst1S mice, and the uncertain timing of infection within the lung. How can the authors be certain that pathology a “stepwise progression”, and smaller “incipient” lesions are earlier lesions, followed by larger lesions? To the above point, could the lesion morphology be influenced by when the lung was seeded with Mtb? From what is shown, inferences can be made about lesion size and bacterial burden, but it is difficult to support the different stage of granuloma development proposed in lines 392-400. The above suggested time course may shed some light onto this question.

There is not enough information on number of replicates to fully evaluate this manuscript. We’re these findings repeated? Also, given the heterogeneity of lesion structures if would be helpful to know how sections were selected for downstream processing. Lastly, how many sections were analyzed per mouse, was there concordance between different sections in the same mouse.

To this end, the authors used semi-quantitative determination of AFB staining was used in most cases, while in others (Figure 3), density was used. How do these determinations compare? Density would seem to be the more objective and quantitative readout between the two. Is there a reason that the authors couldn’t apply this analysis throughout their experiments? If not, I would request that this analysis (or CFU determination, if available) be completed. Also, what was the cutoff for paucibacillary and multibacillary lesions, and how was this determined? If the authors instead used a continuous variable for Mtb density, would they have greater power to identify additional differences? Lastly, Why were lesions with advanced necrosis excluded from some analyses?

On lines 185-186, the claim is made that the bacterial loads in the lungs always exceeded those of other regions. However, there are a number of mice in S2E where the AFB scores for non-lung organs are equal to those of this lung. This should be accurately stated, and more clarity may be shed on this issue by a fully quantitative analysis.

The lung and spleen implantation results are intriguing, and could suggest a lung parenchyma-specific role in driving necrotic lesion formation. Could there be other potential explanations such as the efficiency of neovascularization? Also, could the authors provide a more in depth analysis to lend insight onto the inflammatory mediators that may be driving these processes (potentially informed by the spatial transcriptomics data)? Ifnb production has been sown to inhibit alternatively activated macrophage (27849167). To this point, the transcriptional data from the bone marrow would suggest that this process does not start in the lung. I’m also not sure what figure 7 adds here, why is it important to show in human lungs?

Key details are missing from the organ implantation, and the cited reference for this in the methods section is a preprint. This information should be included in this manuscript to facilitate review. To this end, certain figure panels (Figures 1A, S2E) are duplicated in a preprint from the same authors here: https://www.biorxiv.org/content/10.1101/2025.02.28.640830v1.full (Figures 2A and 2C).

Reviewer #2: No major concerns.

**Part III – Minor Issues: Editorial and Data Presentation Modifications**

Reviewer #1: Figure 1A – Is there a reason this is not shown in the form of a Kaplan meier curve?

Figure 1B – What is this reflective of? Weight gain since when? If this were at the start of infection, I would have expected all mice to have 0% weight gain at day 0.

Figure 1C – Significance indications should be shown here

Line 243 – a decline in the monocyte to macrophage ratio is not shown directly in the figures

A consistent format should be used to display GSEA data, with the inclusion of normalized enrichment score (Similar to figure 5D).

Figure 3E is hard to read, please enlarge text and consider grouping genes into the themes discussed in the text. Also, is the PCA shown?

Figure 3F – how do the authors explain the discrepancy in B cells with 3D?

Fig 4C – Could the authors speculate on what the implications would be of “cloaked macrophages”? They have already been activated, since they express iNOS. Is there previous work showing the consequences of macrophage-macrophage cloaking?

Fig 4E – there is a wide spread in these values (and the values are paired within the same lesion). Is there any relation to the Mtb density?

Lines 274-281 – The text refers to proportions cells and macrophages expressing a certain marker, but only the lesion area is shown. Do the authors have data on cell proportions (e.g. the percentage of Ki67+ macrophages that are iNOS+)?

Reviewer #2: There are minor issues in the manuscript that can be readily addressed:

1. Fig 2B. Do the lymphoid follicles contain mostly plasma cells and macrophages (as suggested in the legend)? This is related to another piece of data presented in Fig. 3F. The levels of plasma cells in the paucibacillary lesions is significantly higher compared to that in multibacillary lesions based on GeoMx spatial transcriptomics studies. It would be desirable to show the markers used to identify plasma cells, particularly in the context of differentiating plasma cells from plasmablasts. Finally, in Fig. 2A, denoting plasma cells with a

sign (arrows, e.g.) would better demonstrate their presence. The authors might also refer to the plasma cell data derived from the spatial transcriptomics study (Fig.3).

2. Fig. 6E, the spleen implant images are absent but described in the figure legend.

3. The manuscript might consider delving a bit more into the discussion concerning post-primary TB, a not well understood entity. Specifically, it would be worthwhile discussing the modified B6.Sst1S-based hematogenous model in the context of post-primary TB, which could possibly differ from the natural post-primary TB scenario, in which the lung tissues that are the locale of bacterial seeding through the bloodstream have been previously infected with M. tuberculosis.

4. The figure legends. In general, the figure legends can include more details so that the materials presented can be followed more easily. For example, information that connect the targeted immune cell subsets and the antibodies used would be desirable. In addition, the reproducibility of the results presented should be described – e.g., how many times had the studies been carried out. The title of Fig. 3 is “Comparative cellular and spatial transcriptomic analysis of paucibacillary and multibacillary TB lesions in B6 and B6.Sst1S mice“. However, the labeling of the results presented in not clear with respect to their relationship to the B6 versus the B6.Sst1S groups.

5. Suppl materials. In Suppl Fig.1A, the mouse strains used were not denoted. In the Suppl Table 2 and Table 3, the superscript “2” of AFB2 was not designated. In Suppl Fig.3 C, the four images of panel C (B6 Pausibacillary) is a bit confusing – specifically, the lymphoid aggregate organization of some of the images shown, particularly the bottom one, are not obvious. But the legend (as well as the text) implies that, while the lymphoid aggregates dissolute in the B6.Sst1S multi-bacillary lesions, they are more prominent in B6 P and B6.Sst1S M tissues. More precise choice of images could clarify this issue. Suppl Fig.6, for clarity and readibility, SPC should be spelled out.

PLOS authors have the option to publish the peer review history of their article (what does this mean?). If published, this will include your full peer review and any attached files.

Reviewer #1: No

Reviewer #2: No

**Figure resubmission:**
---

## [Decision Letter · Decision Letter 1]

24 Sep 2025

Dear Dr. Kramnik,

We are pleased to inform you that your manuscript 'Dysplastic lung repair fosters a tuberculosis-promoting microenvironment through maladaptive macrophage polarization' has been provisionally accepted for publication in PLOS Pathogens.

Best regards,

Anne Jamet

Section Editor

PLOS Pathogens

Sumita Bhaduri-McIntosh

Editor-in-Chief

PLOS Pathogens

orcid.org/0000-0003-2946-9497

Michael Malim

Editor-in-Chief

PLOS Pathogens

orcid.org/0000-0002-7699-2064

Reviewer #1:

Reviewer #2:

Reviewer Comments (if any, and for reference):

Reviewer's Responses to Questions

**Part I - Summary**

Reviewer #1: The authors have addressed most of the concerns well, and this study provides a significant advance to the role of lung microenvironments and systemic immunity in lesion formation.

Reviewer #2: The authors have responded to and clarified the issues raised by this reviewer in the original submission of the manuscript. Clarification of some of the issues raised have provided the manuscript with clarity and more succinct conveyance of the message intended.

**Part II – Major Issues: Key Experiments Required for Acceptance**

Reviewer #1: My only concern is that the author’s rationale supporting the potential broad relevance of this model, including the dose used, and the claim of lesions establishing after adaptive immunity, is only best laid out in the response to reviewers (in the below quoted and preceding paragraphs).

“Taken together these data demonstrate that the development of individual PTB lesions in

our model occurs 6 – 12 wpi, i.e. after the induction of adaptive immunity. Also it is a relatively

rare event – assuming that single Mtb can give rise to a lesions, it is less than 0.0001% of the

inoculum that establishes PTB lesions.”

The authors should include in the main text some discussion and rationale on why 10^6 is a physiologically relevant dose for this model. Their STAR Protocols paper discusses the dose escalation to achieve 100% of mice lesions, but doesn’t directly address whether this is physiologically relevant (one might argue that an inoculum which achieves lesions in 50%, or even less, of mice might more closely mirror what happens clinically). The authors’ point in the response to reviewers that lesions are seeded by 0.0001% of the inoculum is well taken and could be consistent with the infrequent hematogenous seeding that is seen in patients.

Lung pathology should be shown from week 6, and the authors should discuss the timing of lesion in the text (the timing of Mtb in the lymph nodes is only present in a figure legend), to support their claim that lesions do not form early after infection, and are only initiated while adaptive immunity is ongoing.

Reviewer #2: N/A

**Part III – Minor Issues: Editorial and Data Presentation Modifications**

Reviewer #1: Rationale should be provided in the text for excluding necrotic lesions

The data from B6.Sst1S mice in Figures 1A and S2E is presented in the authors' recently published Star Protocols paper. Consider citing as such.

Reviewer #2: N/A

PLOS authors have the option to publish the peer review history of their article (what does this mean?). If published, this will include your full peer review and any attached files.

Reviewer #1: No

Reviewer #2: No

---

## [Editor Report · Acceptance letter]

Dear Dr. Kramnik,

We are delighted to inform you that your manuscript, "Dysplastic lung repair fosters a tuberculosis-promoting microenvironment through maladaptive macrophage polarization," has been formally accepted for publication in PLOS Pathogens.

Best regards,

Sumita Bhaduri-McIntosh

Editor-in-Chief

PLOS Pathogens

orcid.org/0000-0003-2946-9497

Michael Malim

Editor-in-Chief

PLOS Pathogens

orcid.org/0000-0002-7699-2064